# Identification of a dual orange/far-red and blue light photoreceptor from an oceanic green picoplankton

Yuko Makita[1,13], Shigekatsu Suzuki[2,13], Keiji Fushimi[3,4,5,13], Setsuko Shimada[1,13], Aya Suehisa[1], Manami Hirata[1], Tomoko Kuriyama[1], Yukio Kurihara[1], Hidefumi Hamasaki[1,6], Emiko Okubo-Kurihara[1], Kazutoshi Yoshitake[7], Tsuyoshi Watanabe[8], Masaaki Sakuta [9], Takashi Gojobori[10], Tomoko Sakami[11], Rei Narikawa [3,4,5,12], Haruyo Yamaguchi [2], Masanobu Kawachi[2] & Minami Matsui [1,6✉]

Photoreceptors are conserved in green algae to land plants and regulate various developmental stages. In the ocean, blue light penetrates deeper than red light, and blue-light sensing is key to adapting to marine environments. Here, a search for blue-light photoreceptors in the marine metagenome uncover a chimeric gene composed of a phytochrome and a cryptochrome (*Dualchrome1*, *DUC1*) in a prasinophyte, *Pycnococcus provasolii*. DUC1 detects light within the orange/far-red and blue spectra, and acts as a dual photoreceptor. Analyses of its genome reveal the possible mechanisms of light adaptation. Genes for the light-harvesting complex (LHC) are duplicated and transcriptionally regulated under monochromatic orange/blue light, suggesting *P. provasolii* has acquired environmental adaptability to a wide range of light spectra and intensities.

[1] Synthetic Genomics Research Group, RIKEN Center for Sustainable Resource Science, Yokohama, Japan. [2] Biodiversity Division, National Institute for Environmental Studies, Tsukuba, Japan. [3] Graduate School of Integrated Science and Technology, Shizuoka University, Shizuoka, Japan. [4] Research Institute of Green Science and Technology, Shizuoka University, Shizuoka, Japan. [5] Core Research for Evolutional Science and Technology, Japan Science and Technology Agency, Saitama, Japan. [6] Yokohama City University, Kihara Institute for Biological Research, Yokohama, Japan. [7] Graduate School of Agricultural and Life Sciences, The University of Tokyo, Tokyo, Japan. [8] Fisheries Resources Institute, Japan Fisheries Research and Education Agency, Kushiro, Hokkaido, Japan. [9] Department of Biological Sciences, Ochanomizu University, Tokyo, Japan. [10] Computational Bioscience Research Center, King Abdullah University of Science and Technology, Thuwal, Kingdom of Saudi Arabia. [11] Fisheries Resources Institute, Japan Fisheries Research and Education Agency, Minami-ise, Mie, Japan. [12] Department of Biological Sciences, Graduate School of Science, Tokyo Metropolitan University, Tokyo, Japan. [13] These authors contributed equally: Yuko Makita, Shigekatsu Suzuki, Keiji Fushimi, Setsuko Shimada. ✉email: minami@riken.jp

Photosynthetic organisms utilize various wavelengths of light, not only as sources of energy but also as clues to assess their environmental conditions. Blue light penetrates deeper into the ocean, whereas red light is absorbed and immediately decreases at the surface. Oceanic red algae possess blue-light receptor cryptochromes (CRYs) but not red-light receptor phytochromes (PHYs)[1]. Similarly, most chlorophytes have CRYs but fewer have PHYs. PHYs are bilin-containing photoreceptors for the red/far-red-light response. Interestingly, algal PHYs are not limited to red and far-red responses. Instead, different algal PHYs can sense orange, green, and even blue light[2]. They have the ability to photosense between red-absorbing Pr and far-red-absorbing Pfr, and this conformational change enables interactions with signaling partners[3]. CRY is a photolyase-like flavoprotein and widely distributed in bacteria, fungi, animals and plants.

In 2012–2014, large-scale metagenome analyses were performed in Sendai Bay, Japan, and the western subarctic Pacific Ocean after the Great East Japan Earthquake to monitor its effects on the ocean (http://marine-meta.healthscience.sci.waseda.ac.jp/crest/metacrest/graphs/). These metagenome analyses targeted eukaryotic marine microorganisms as well as bacteria. Blue-light sensing is key to adapting to marine environments. From a search for CRYs in the marine metagenomic data, we found a chimeric photoreceptor, designated as Dualchrome1 (DUC1), consisting of a two-domain fusion of PHY and CRY. We found that DUC1 originated from a prasinophyte alga, Pycnococcus provasolii.

P. provasolii is a marine coccoid alga in Pseudoscourfieldiales, Pycnococcaceae (or prasinophyte clade V[4]) and was originally discovered in the pycnocline[5]. P. provasolii is classified in Chlorophyta, which is a sister group of the Streptophyta in Viridiplantae. Chlorophyta contain three major algal groups, Ulvophyceae, Trebouxiophyceae and Chlorophyceae (UTC clade), and "prasinophytes", which have several characteristics considered to represent the last common ancestor of Viridiplantae. Prasinophytes mainly inhabit marine environments and are dominant algae under various light qualities and intensities[6]. Thus, prasinophytes are key to understanding the diversity and evolutionary history of the light response system in the Viridiplantae.

Environmental DNA research shows P. provasolii lives at depths in the range of 0–100 m and in varying regions of the marine environment[6,7]. It also has a unique pigment composition (prasinoxanthin and Magnesium 2,4-divinylpheoporphyrin $a_5$ monomethyl ester) and an ability to adapt to the spectral quality (blue and blue-violet) and low fluxes of light found in the deep euphotic zone of the open sea[5,8].

We unveil here with DUC's ability to detect a wide range of the light spectrum (orange to far-red for PHY and UV to blue for CRY) and undergo dual photoconversions at both the PHY and CRY regions. We sequence the genome of P. provasolii and examine its light-associated features. These findings will help to understand the evolutionary diversity of photoreceptors in algae and explain the environmental adaptation and success of P. provasolii.

## Results

**Identification of photoreceptor DUC1 from marine metagenome data.** Since CRYs have been reported in several chlorophytes, we searched the marine metagenome data with cryptochrome PHR and FAD as baits to target CRY genes. We called 542 assembled metagenome sequences with the PHR, FAD and Rossmann-like_a/b/a_fold domains, and used a combination of these three domains as a hallmark for CRY candidates. Among the candidate genes identified, we found one with PHY-like sequence at its N-terminal region similar to Arabidopsis

phytochrome B (PHYB) (Fig. 1). This gene has no introns and can encode angiosperm PHY domains at its N-terminus and CRY domains at its C-terminus (Fig. 1a).

In metagenome data from four collection points, fragments of this gene were detected in open sea areas (C12 and A21 points) at both the Sendai Bay and A-line sampling stations (Supplementary Fig. 1a). To find the host plankton or its relative, we searched for a similar sequence in MMETSP (The Marine Microbial Eukaryote Transcriptome Sequencing Project). We found transcriptome fragments that matched with 100% identity to this gene, although we did not isolate the boundary sequence between the PHY and CRY regions. We identified the host plankton as a marine prasinophyte alga, P. provasolii. Fortunately, a culture strain of it was stocked in the NIES collection as P. provasolii NIES-2893 (Fig. 1b). Metagenome data also supported the fact that there was enrichment of this species in the subsurface chlorophyll maximum layer of stations C12 and A21 (Supplementary Fig. 1, Supplementary Note 1, Supplementary Method 1). After amplification of this gene from NIES-2893 by RT-PCR, we finally concluded that it does have both the PHY and CRY domains (5073 bp, 183 kDa protein) and designated it as Dualchrome1 or DUC1 (Supplementary Fig. 2a).

Phylogenetic analysis using the P. provasolii PHY (PpPHY) and CRY (PpCRY) domains showed they branched with those of other chlorophytes (Supplementary Figs. 3 and 4), which is consistent with the phylogenetic position of this species based on the 18S rRNA gene. Additionally, we found that three other strains of P. provasolii and Pseudoscourfieldia marina NIES-1419, which is a close relative of P. provasolii, also possess the DUC1 gene. P. marina DUC1 showed 98.2% amino acid identity with the P. provasolii DUC1 (PpDUC1) (Supplementary Fig. 2b, c).

**PpDUC1 senses blue, orange, and far-red light.** To verify whether DUC1 possesses photosensing activity, we expressed the PHY (PpPHY, 70.2 kDa, amino acid positions 1–662 in PpDUC1) and CRY (PpCRY, 72.2 kDa, amino acid positions 1039–1690 in PpDUC1) regions with His-tag and GST-tag, separately (Fig. 2a).

PHYs have a Cys residue either within the GAF domain or in the N-terminal loop region to covalently bind to the bilin chromophore (Supplementary Fig. 5a). The bilin chromophore shows light-induced Z/E isomerization that triggers a reversible photocycle between the dark state (or ground state) and the photoproduct state (or excited state) of the PHYs (Supplementary Fig. 5a)[9]. The GAF Cys ligates to C3$^1$ of phycocyanobilin (PCB) or phytochromobilin (PΦB), whereas the N-terminal Cys ligates to C3$^2$ of biliverdin IXα (BV). As the PpPHY possesses the GAF Cys residue but not the N-terminal Cys, the binding chromophore is likely to be PCB or PΦB. Furthermore, we detected a bilin reductase homolog from the P. provasolii NIES-2893 genome (described later), which belongs to the PcyA cluster, producing PCB from BV, but not to the HY2 cluster, producing PΦB from BV (Supplementary Fig. 6a, b)[10]. Taken together, we presumed that the native chromophore for PpPHY is PCB and not PΦB. Therefore, we expressed PpPHY in E. coli harboring the PCB-synthetic system (C41_pKT271)[11]. The purified PpPHY covalently bound PCB (Supplementary Figs. 5b and 6c) and showed reversible photoconversion between an orange-absorbing (Po) form ($\lambda_{max}$, 612 nm) in the dark state and a far-red-absorbing (Pfr) form ($\lambda_{max}$, 702 nm) in the photoproduct state (Fig. 2b and Supplementary Table 1). Dark reversion from the photoproduct state to the dark state was not seen (Supplementary Fig. 7a). Furthermore, we observed that PΦB but not BV could be efficiently incorporated into the apo-PpPHY to show reversible photoconversion by using the PΦB- and BV-synthetic systems, which is consistent with previous studies (Supplementary

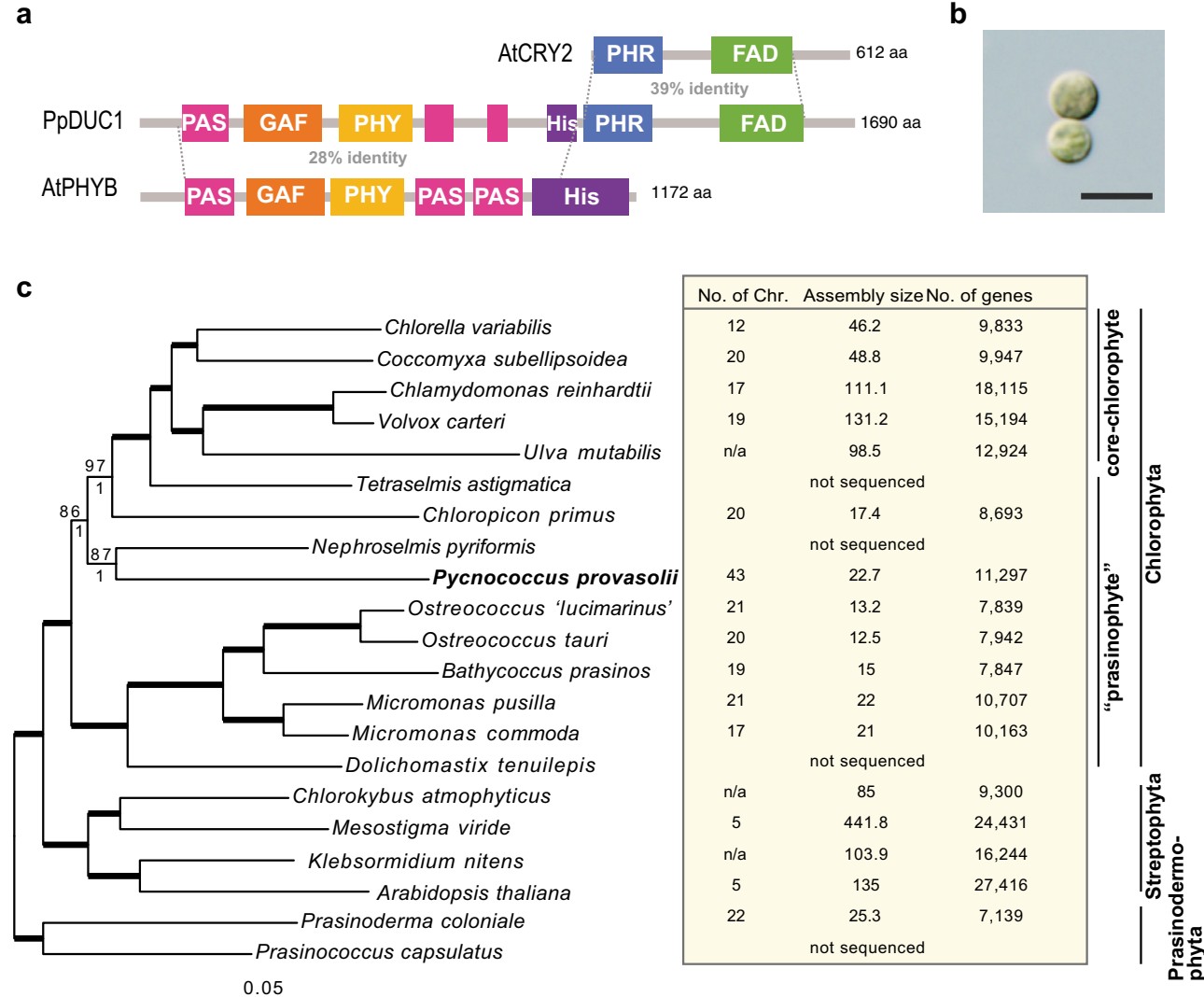

**Fig. 1 Chimeric photoreceptor PpDUC1 and phylogeny of its host *P. provasolii*. a** Domain conservation between PpDUC1 and *Arabidopsis* CRY2 and PHYB. Domain structure of PpDUC1 constructed with PAS: Per-Arnt-Sim, GAF: cGMP-specific phosphodiesterases, adenylyl cyclases and FhlA. PHY: phytochrome, His: histidine kinase domain, PHR: photolyase-homologous region, and FAD: flavin adenine dinucleotide. **b** Light microscopic image of coccoid cells of *P. provasolii* NIES-2893. Scale bar = 5 μm. Observations were repeated more than three times and representative cells are shown. **c** Phylogeny of *P. provasolii* and genome statistics of the main lineages of Viridiplantae. Maximum likelihood (ML) tree of 105 orthologous proteins. The dataset was composed of 41,023 amino acids from 21 species that have genomes available, including *P. provasolii*. Bootstrap percentages (BPs) and Bayesian posterior probabilities (BPP) are shown above and below the lines, respectively. Bold lines show BP = 100 and BPP = 1.00.

Fig. 6c–f). Taking the presence of the PcyA homolog into consideration, it is likely that PpPHY reversibly senses orange and far-red light via PCB incorporation in vivo.

CRYs non-covalently bind to a flavin chromophore, in many cases, flavin adenine dinucleotide (FAD) (Supplementary Fig. 5c). Some redox state forms of FAD, such as oxidized FAD ($FAD_{OX}$), anion radical FAD ($FAD^{\bullet-}$), neutral radical FAD ($FADH^{\bullet}$) and reduced FAD ($FAD_{red}H^{-}$), are observed in the photocycle of CRYs (Supplementary Fig. 5c)[12]. As the PpCRY accumulates as insoluble inclusion bodies in the normal expression system, we expressed PpCRY in *E. coli* harboring the chaperon co-expression system (C41_pG-KJE8)[13]. The purified PpCRY non-covalently bound FAD (Supplementary Figs. 5d and 6c). The protein showed a blue-absorbing (Pb) form ($\lambda_{max}$, 447 nm) before light irradiation (Fig. 2c and Supplementary Table 1). The spectral form in the dark state was similar to that of the $FAD_{OX}$-bound CRYs[12]. Blue-light irradiation resulted in conversion to a UV-absorbing (Puv) form ($\lambda_{max}$, 366 and 399 nm) (Fig. 2c and Supplementary Table 1). The spectral form in

the photoproduct state was similar to that of the $FAD^{\bullet-}$-bound CRYs[12]. Although we further irradiated the protein with UV-to-blue light, no spectral change was observed. Instead, Puv-to-Pb dark reversion occurred (Supplementary Fig. 7b). To conclude, PpCRY showed photoconversion from the $FAD_{OX}$-bound form to the $FAD^{\bullet-}$-bound one, and the reverse reaction occurred in the dark. This is similar to the photocycle of dCRY, a photoreceptor for circadian clock regulation in *Drosophila melanogaster*[14], rather than CraCRY from the green algae *Chlamydomonas reinhardtii*[15] and AtCRY1 and 2 from the land plant *Arabidopsis thaliana*[16].

These results suggest that PpDUC1 is a broadband light sensor that can detect long-wavelength light (i.e., orange to far-red) in the PpPHY region and short-wavelength light (i.e., UV to blue) in the PpCRY region.

**PpDUC1 mainly localizes at nucleus in tobacco leaves**. It is reported that light conditions change the intracellular localization of photoreceptors in higher plants and *Micromonas pusilla*

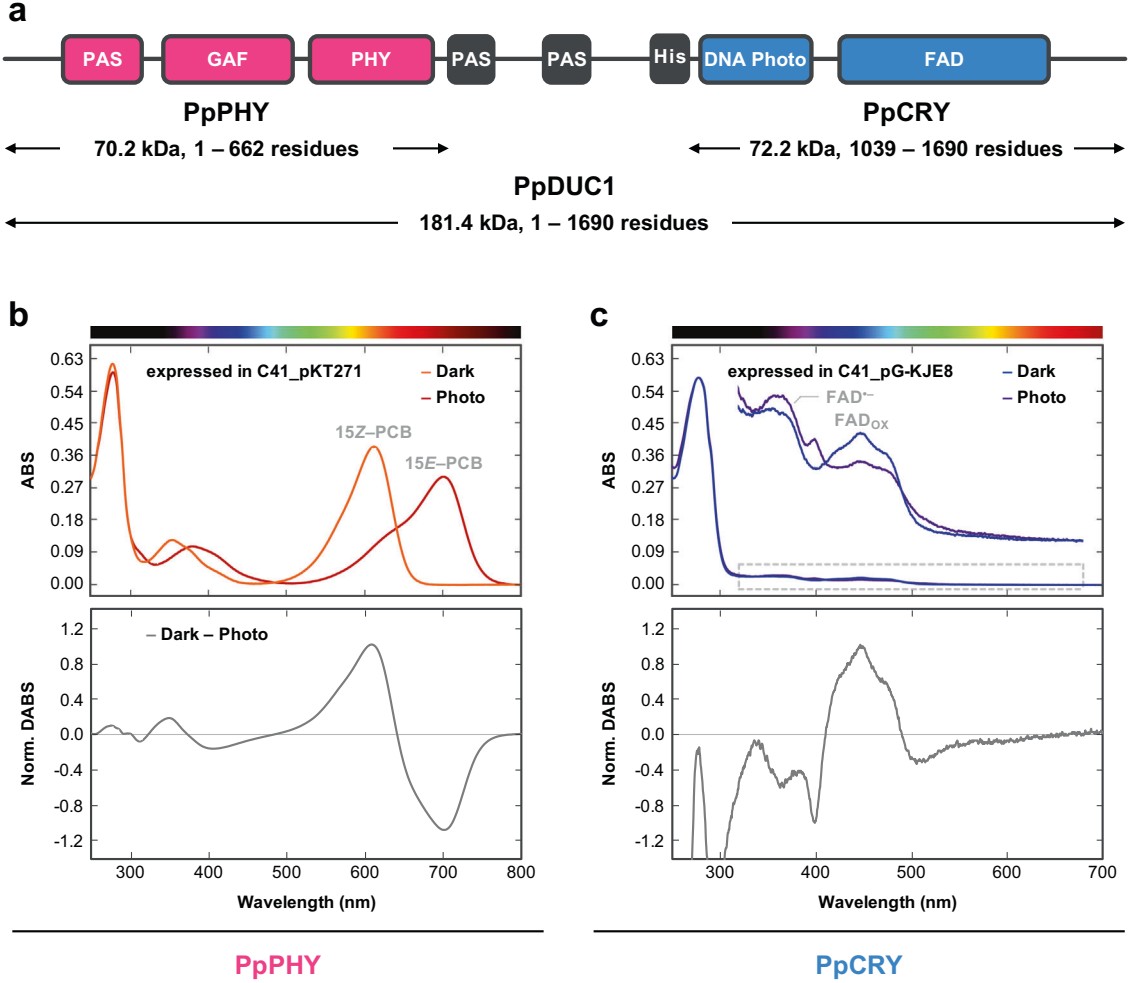

**Fig. 2 Photoconversion of PpPHY and PpCRY. a** Domain structure of PpDUC. PpPHY (magenta) and PpCRY (cyan) regions are shown with their molecular sizes and amino acid positions. **b** Absorption spectra of the dark state (15Z–PCB, Po form) and the photoproduct state (15E–PCB, Pfr form) of PpPHY expressed in C41_pKT271 (harboring the PCB-synthetic system). The normalized difference spectrum (dark state—photoproduct state) was calculated from these absorption spectra. **c** Absorption spectra of the dark state (FAD$_{OX}$, Pb form) and the photoproduct state (FAD$^{\cdot-}$, Puv form) of PpCRY expressed in C41_pG-KJE8 (harboring the chaperone expression system). The normalized difference spectrum (dark state— photoproduct state) was calculated from these absorption spectra. The absorption maxima of PpPHY and PpCRY are reported in Supplementary Table 1. Assignment of chromophores incorporated in PpPHY and PpCRY was performed by comparison with each standard, which are reported in Supplementary Fig. 5.

CCMP1545, a marine microalgae[17]. We examined PpDUC1 localization using tobacco (*Nicotiana benthamiana*) cells. PpDUC1::GFP, as well as the PHY region of PpDUC1 (PpPHY:: GFP) and the CRY region of PpDUC1 (PpCRY::GFP), were introduced into tobacco cells (Fig. 3a). Expression of the GFP-fused proteins was confirmed by protein-blot analysis (Fig. 3b). *Arabidopsis* phyB is reported to localize in the nucleus after light irradiation[18] but, although PpPHY::GFP contains all the PHY sequences homologous to AtPHYB, it localized in the cytoplasm (Fig. 3c). *Arabidopsis* cry2 localizes in the nucleus[19] and PpCRY:: GFP was found mainly here with a weak signal in the cytoplasm. PpDUC1::GFP exhibited mostly nuclear localization. We examined subcellular localization of PpDUC1::GFP in the dark and then on transfer to the light. In the darkness, PpDUC1::GFP also mostly localized in the nucleus, and, after white light irradiation, it did not show a clear change of localization (Supplementary Fig. 8).

**P. provasolii possesses a unique gene repertoire for adaptation to various light conditions**. We sequenced the complete genome of *P. provasolii* NIES-2893. The reads were assembled into 43 scaffolds without gaps and the scaffolds likely correspond to chromosomes (Supplementary Fig. 9, Supplementary Note 2, Supplementary Method 2). The genome size of *P. provasolii* is 22.7 Mbp, which is similar to other prasinophytes (12.5–22.0 Mbp) (Fig. 1c and Supplementary Method 3–5). We predicted and annotated 11,297 genes and evaluated gene annotation completeness with BUSCO. The results showed 89.1% (270/303) complete, 4.8% (16/303) fragmented and only 5.6% (17/303) missing BUSCOs. Among the seven prasinophytes that have their genomes determined, we identified 3996 conserved orthogroups (47.8% of *P. provasolii* genes) and a large number of unique genes (3636 orthogroups, 43.5% of the total) (Supplementary Fig. 9c).

With this genome sequence, we predicted five *CRYs* and one *DUC1* but there is no *PHY* in the genome (Table 1). Of the *CRY* genes, two and the *PpCRY* region of *PpDUC1* belong to the plant *CRYs* (*pCRY*[15]). These three genes form a monophyly and branch at the basal position of those of other chlorophytes (Supplementary Fig. 3). Chlorophyte-specific *CRYs* are a sister group of the streptophyte *pCRYs*, including *AtCRY1* and *AtCRY2*. The *PpPHY* region is a monophyly with other prasinophytes and streptophytes (Supplementary Fig. 4). In the prasinophytes, two genomes of mamiellophyceans lack genes for *pCRY* and *PHY* (Fig. 4).

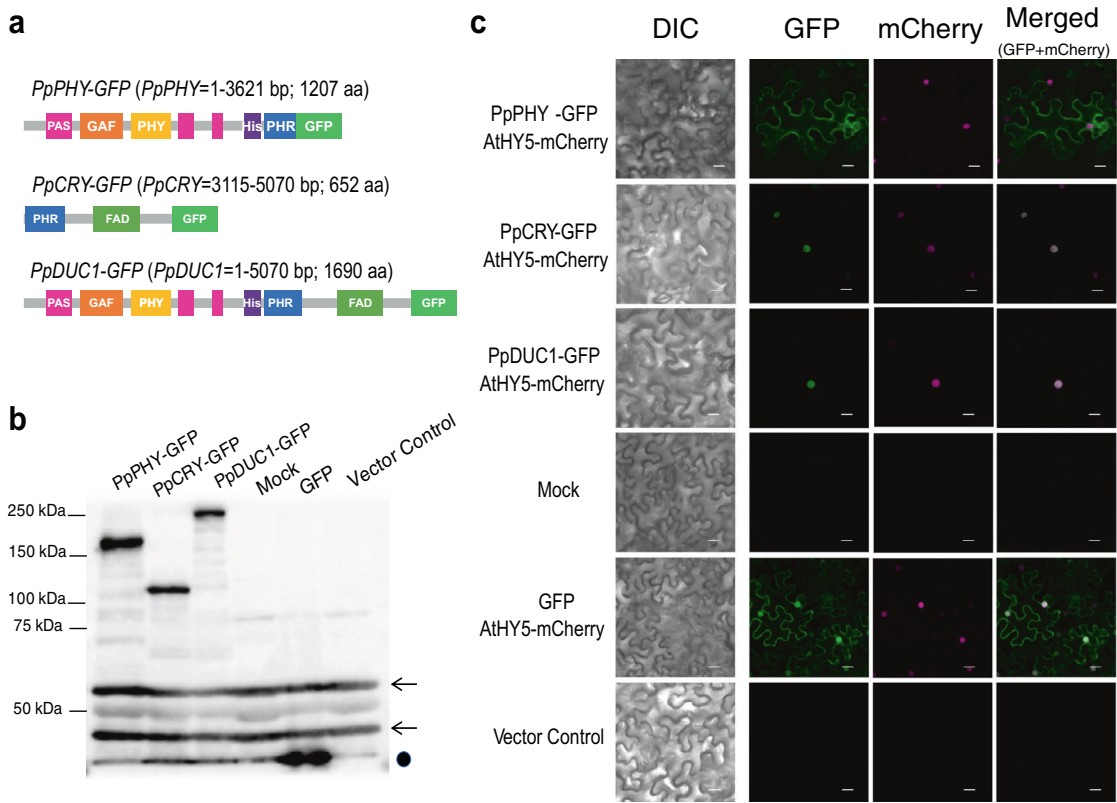

**Fig. 3 Subcellular localization of PpDUC1 in tobacco leaves. a** Schematic diagrams of the domains in the PpPHY, PpCRY and PpDUC1 constructs used in the *N. benthamiana* leaf injection assay. **b** Immunoblot images of protein extract from *N. benthamiana* leaf tissue. Each protein was detected using an α-GFP antibody. Mock, non-injected leaf. Black arrows denote non-specific bands. Dot indicates the dye front. Experiments were repeated two times, and the results of one representative experiment are shown. **c** Leaf injection assay in *N. benthamiana* pavement cells transiently expressing PpPHY-GFP, PpCRY-GFP and PpDUC1-GFP with HY5-mCherry to indicate the nucleus. Observations were performed three times with similar results. DIC, differential interference contrast images; GFP, GFP fluorescence images; Merged, merged images of GFP and mCherry fluorescence images. Scale bar = 20 μm. The source data underlying Fig. 3b are provided as a Source Data file.

The predicted genes also suggest that several important photoreceptors and light signal transduction genes in land plants are conserved. We detected *PHOT*, *COP1*, *CK2alpha*, some subunits of signalosome and *HY5* homologs but not those for *PIF*, *SPA1*, *SPA3*, *FHY1*, *FHY3*, *CIB1*, or *BIC* for light signal transduction. The deduced amino acids sequence of *LAF1* showed 44% identity with Arabidopsis LAF1 although this protein is suggested as an angiosperm-specific protein[20]. We also confirmed the conservation of light signal transduction genes in 13 chlorophyte genomes, but *PIF*, *CIB1* and *BIC* were not found in any (Table 1 and Supplementary Table 2).

For photosynthesis-related genes, its genome encoded a large gene family for a unique prasinophyte-specific light-harvesting complex (*Lhcp*). *P. provasolii* possesses a relatively large number of 16 *Lhcp* genes, and 10 of these are clustered in specific regions of chromosome 29 (Supplementary Fig. 10a). These 10 genes had almost the same sequences and were monophyletic (Supplementary Fig. 10a and Supplementary Method 6). Similar duplications of *lhcb* genes are observed in Mamiellophyceae[21], however, these have occurred independently to *P. provasolii*.

**P. provasolii responds to orange, blue, and far-red light for gene expression**. We performed RNA-seq analysis to know how *P. provasolii* responds to monochromatic light after dark acclimatization. The expression of 1,503 genes and 1,147 genes changed after orange- and blue-light irradiation, respectively (Fig. 5a, b left). Through gene ontology (GO) enrichment analysis of the common differentially expressed genes (DEGs) in orange-

and blue-light conditions, photosynthesis and light-harvesting proteins were identified (Fig. 5c). To distinguish the effect of photosynthesis, we applied the photosynthesis inhibitor DCMU (3-(3,4-dichlorophenyl)-1,1-dimethylurea), which inhibits the electron flow from photosystem II to plastoquinone. As a result, the expression of many photosynthesis-related genes, including *LHC*s were not induced under orange light with DCMU (Supplementary Figs. 10b and 11). The number of DEGs with DCMU are shown in Fig. 5a, b right. About 69% of orange light-controlled genes are affected by DCMU (Fig. 5a). Blue light also gave DEGs although the affected number was smaller compared to orange light. There were almost no differences in dark conditions with/without DCMU (Supplementary Fig. 11a).

We summarized the DEGs induced by DCMU shown in Fig. 5b. In total 1,108 genes are indicated with their normalized expression values and ratios between two light conditions. The 119 DEGs seen in OBF (orange, Blue, and Far-red) light contain genes for elongation factor 2 kinase, Mcm2-7 hexameric complex component, DNA replication licensing factor, mcm4 component, and DNA replication protein psf2. Among these DEGs, we observed 45 genes that showed higher expression (>1.5x) in blue light compared to orange light and two genes with lower expression in blue light compared to orange light. Twenty-one genes showed higher expression in blue light compared to far-red light and 31 genes showed lower expression in orange light compared to far-red light.

Among the DEGs under monochromatic light conditions with DCMU was *DUC1*, which was significantly expressed in

**Table 1 Homologs of Arabidopsis photoreceptors and signal-transducing factors in chlorophytes with known genomes.**

| | Organism | PHY | Plant CRY | Plant-like CRY (Chlorophyte-specific CRY) | CRY-DASH | PIF | COP1 | SPA | CK2 alpha | HY5 | BIC | CIB1 | FT | PHOT | UVR8 | CSN1-8 | EIN3 | EBF1 | FHY1,3 | LAF1 | DET1 | DDB1a | CUL4 |
|---|---|---|---|---|---|---|---|---|---|---|---|---|---|---|---|---|---|---|---|---|---|---|---|
| Angiospermae | Arabidopsis thaliana | 4 | 2 | - | 1 | 8 | 1 | 4 | 4 | 1 | 2 | 1 | 1 | 2 | 1 | 10 | 1 | 1 | 2 | 1 | 1 | 1 | 1 |
| Prasinophytes | Pycnococcus provasolii | DUC1 | 2 + DUC1 | 2 | 2 | - | 1 | 1 | 1 | 1 | - | 1 | - | - | - | 7 | - | - | - | 1 | 1 | 1 | 1 |
| | Bathycoccus prasinos | - | - | - | - | - | 1 | 1 | 1 | - | - | - | - | - | - | 8 | - | - | - | 1 | 1 | 1 | 1 |
| | Ostreococcus 'lucimarinus' | - | - | 2 | - | - | 1 | 1 | 1 | - | - | - | - | - | - | 10 | - | - | - | 1 | 1 | 1 | 1 |
| | Ostreococcus sp. RCC809 | - | - | - | - | - | 1 | 1 | 1 | - | - | - | - | - | - | 10 | - | - | - | 1 | 1 | 1 | 1 |
| | Ostreococcus tauri | - | - | - | - | - | 1 | 1 | 1 | - | - | - | - | - | - | 8 | - | - | - | 1 | 1 | 1 | 1 |
| | Micromonas commoda | - | 2 | 2 | - | - | 1 | 1 | 1 | - | - | - | - | - | - | 12 | - | - | - | 1 | 1 | 1 | 1 |
| | Micromonas pusilla | 1 | 2 | 2 | - | - | 1 | 1 | 1 | - | - | - | - | - | - | 11 | - | - | - | 1 | 1 | 1 | 1 |
| | Chloropicon primus | - | 1 | 1 | 2 | - | 1 | - | 3 | - | - | - | - | - | 2 | 12 | - | - | - | 1 | 1 | 1 | 1 |
| UTC clade | Chlorella variabilis NC64A | - | - | 1 | 2 | - | - | 1 | 1 | 1 | - | - | - | - | 2 | 12 | - | - | - | 1 | 1 | 1 | 1 |
| | Coccomyxa subellipsoidea | - | - | 1 | - | - | 1 | - | 3 | - | - | - | - | - | 1 | 12 | - | - | - | 1 | 1 | 1 | 1 |
| | Volvox carteri | 1 | - | 2 | 2 | - | 1 | 1 | 3 | - | - | - | - | 1 | 1 | 12 | - | - | - | 1 | 1 | 1 | 1 |
| | Chlamydomonas reinhardtii | 1 | - | 2 | 2 | - | 1 | 1 | 3 | 1 | - | - | - | 1 | 2 | 13 | - | - | - | 1 | 1 | 1 | 1 |
| | Ulva mutabilis | 1 | 1 | 1 | 1 | - | 1 | 1 | 1 | 1 | - | - | 1 | 1 | 1 | 12 | - | - | - | 1 | 1 | 1 | 1 |

"-" indicates that there is no homologous gene in the genome annotation. Detailed gene names and criteria of homologous genes are provided as a Source Data file. COP1 constitutive photomorphogenic 1, SPA suppressor of phytochrome A, CK2 alpha casein kinase 2 alpha, HY5 elongated hypocotyl 5, BIC B light inhibitors of cryptochromes, CIB1 cryptochrome-interacting basic-helix-loop-helix 1, FT flowering locus T, PHOT phototropin, UVR8 UV resistance locus 8, CSN cop9 signalosome subunit, EIN3 ethylene-insensitive 3, EBF1 EIN3-binding F box protein 1, FHY far-red elongated hypocotyl, LAF1 long after far-red light 1, DET1 de-etiolated 1, DDB1a damaged DNA binding protein 1A, CUL4 cullin 4.

blue-light conditions (Fig. 5d). The *HY5* homolog was also significantly expressed in blue and orange light with DCMU. Homologs of other light-signaling genes are listed in Fig. 5d.

Using real-time PCR, we found that the expression of two *lhcp* genes is induced mostly by orange and blue light, and DCMU treatment reduced their expression (Supplementary Fig. 12 and Supplementary Method 7). *DUC1* expression was also induced by all light conditions and expression caused by orange and blue light was enhanced by DCMU treatment. These real-time PCR results were represented by RNA-Seq analysis.

## Discussion

In this research, we have found a bifunctional photoreceptor, PpDUC1, composed of a fusion of PHY and CRY. In terms of evolution, it is often speculated that different domains of one organism's protein are encoded by separate genes in another and this has been used successfully to speculate about direct physical interaction or indirect functional association[22]. It is reported that phyB and cry2 physically interact to transduce light signals for controlling flowering time in *Arabidopsis*[23]. Our discovery of PpDUC1 indicates that PHY and CRY interact actively to enable proper perception of light signals. Another example is Neochrome, which is found in ferns[24], that is composed of a PHY domain and a PHOT domain. This chimeric protein also supports the idea that there is dynamic interaction of photoreceptors[25].

From RNA-Seq analysis we found *P. provasolii* responds to orange, blue and far-red light and that 1,964 genes are expressed under orange and blue light. DCMU treatment reduced the number of DEGs to 1,094, orange DEGs being reduced from 1,503 to 471. Most of the genes whose expression was canceled by DCMU are genes involved in photosynthesis (Supplementary Fig. 11). Since DUC1 can sense orange, blue and far-red lights, some of the 119 DEGs of OBF light may be controlled by DUC1 (Fig. 5b). Interestingly 45 out of these 119 genes showed higher expression in blue compared to orange light. On the other hand, among 155 DEGs of orange and blue lights only 6 genes showed higher expression in blue compared to orange light (Fig. 5b). These 45 genes may be controlled by other cryptochromes enabling them to achieve their higher expression. *DUC1*, *Plant-like CRY*, *CRY-dash* and *HY5* were all induced by blue light with DCMU treatment. Inhibition of photosynthesis may control expression of these genes (Fig. 5d).

In tobacco cells, PpDUC1 mostly localizes in the nucleus under white light (Fig. 3f) while the PpPHY domain is mainly localized in the cytoplasm. Although this investigation was done in a heterologous system, PpPHY's intracellular localization did not change under different light conditions.

This may explain the results of the complementation assay that shows PpDUC1 does not complement *phyB* in respect of hypocotyl length (Supplementary Figs. 13 and 14, Supplementary Method 8 and 9). Plant and *M. pusilla* phytochromes translocate from the cytoplasm to the nucleus under light irradiation[17]. Plant phytochrome is known to transduce its photoactivated signals through interaction with PIF protein in the nucleus and finally HY5 controls light-inducible gene expression[26]. We did not observe light-dependent intracellular PpDUC1 translocation nor its enrichment in the nucleus by light in tobacco cells (Supplementary Fig. 8). Also, there is no PIF homolog in the *P. provasolii* genome (Table 1). Further analysis is needed to understand how DUC1 transduces light signals to control gene expression in *P. provasolii*.

In this study, we have revealed that PpPHY shows Po/Pfr photocycle, which is similar to the PHY molecules derived from other prasinophyte species, indicative of the same origin[2]. It is

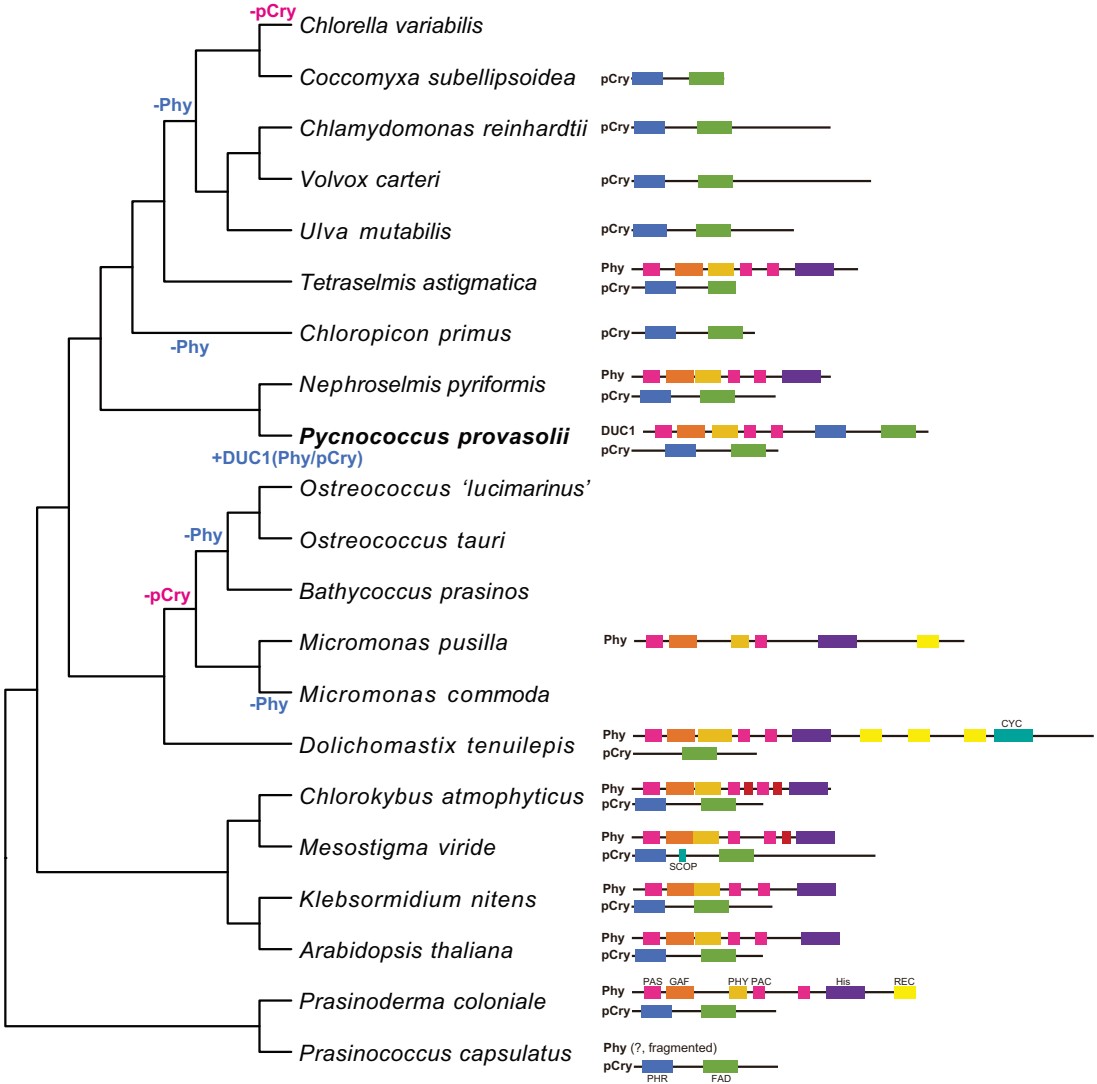

**Fig. 4 Evolutionary scenario of PHY and pCRY in chlorophytes.** The phylogenetic relationships of representative chlorophytes and the domain structures of their PHYs and pCRYs are shown. The tree topology is based on this study. The NCBI-nr database or MMETSP assemblies were searched for PHYs and pCRYs. Each domain is depicted by a different color.

well known in cyanobacteriochrome photoreceptors, distant relatives of the PHYs, that the trapped geometry of the rotating ring D is crucial for blue-shifted absorption[27]. Residues unique to prasinophyte PHY molecules would be crucial for such a twist of ring D. Notably, two Tyr residues conserved among the plant and cyanobacterial PHYs that hold ring D are replaced with Phe, Met or Trp residues in the prasinophyte PHYs (Supplementary Fig. 15 and Supplementary Method 10)[28], which may contribute to holding ring D in the trapped geometry, resulting in the absorption of blue-shifted orange light in the dark state Po form.

Many green algae share genes for *PHY*[9] and *pCRY*[29]. Of the prasinophytes, *Tetraselmis*, *Nephroselmis*, *Micromonas*, *Dolichomactix*, and *Prasinoderma* possess genes for functional *PHY*[1,2]. However, the genomes of *Chloropicon*, *Ostreococcus*, *Bathycoccus*, and *M. commoda* lack any *PHY* genes (Table 1 and Fig. 4). These PHYs are monophyletic and the tree topology coincides with the species tree (Fig. 1c and Supplementary Fig. 4), suggesting that *PHY* may have disappeared multiple times, independently[9]. Our phylogenetic analysis inclusive of DUC1 supports the idea that the last common ancestor of the Archaeplastida had a phytochrome[17]. In contrast, *pCRY* is widely shared in chlorophytes and streptophytes[15], and only the genomes of

Mamiellales and *Chlorella variabilis* lack *pCRY* (Supplementary Fig. 3). In evolutionary terms, *PpDUC1* was not found in other green algae except for *P. provasolii* and *Pseudoscourfieldia*, and *P. provasolii* is sister to *N. pyriformis*, which possesses *PHY* and *pCRY*, which suggest that *PpDUC1* may have been acquired in an ancestor of *Pycnococcus* (and *Pseudoscourfieldia*). As the domain structure of PpDUC1 is similar to the Phy and Cry of *N. pyriformis* (Fig. 4), *PpDUC1* may have been generated via the fusion of these genes. Recent fusion is suggested by the remaining original GC%, i.e., the PHY region has higher GC% than the pCRY region in PpDUC1 (PpPHY: 62.2% and PpCRY: 59.5%).

Sensitivity to weak light is essential for marine algae. *P. provasolii* also has a unique pigment composition (prasinoxanthin and Magnesium 2,4-divinylpheoporphyrin $a_5$ monomethyl ester) and an ability to adapt to spectral quality (blue and blue-violet) and low-light intensity[5,8]. Under high-light radiation, algae and land plants degenerate antenna complexes (LHCs and pigments) to decrease the absorbance of excess energy and prevent photodamage[30]. The degradation of Chl triggers degradation of LHCs[31]. Interestingly, *LHC* induction by orange and blue light is strongly reduced by DCMU treatment but *ELIP* expression is induced by DCMU treatment. ELIPs are members of LHCs that

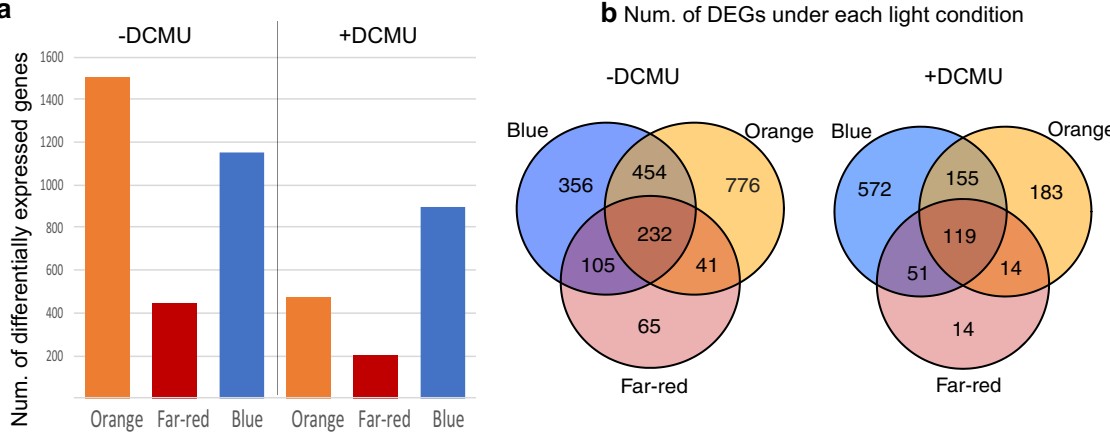

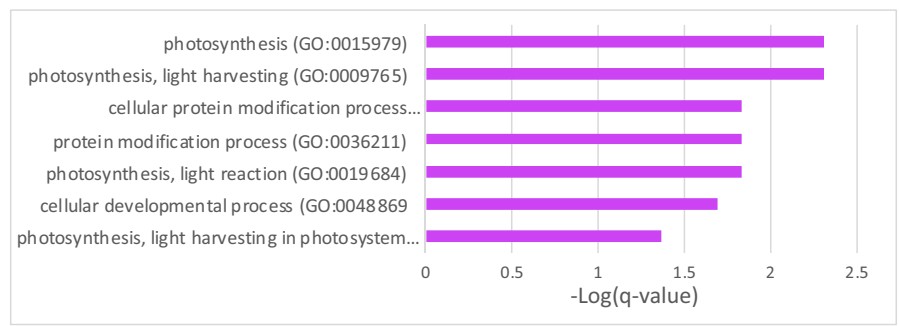

**c**. Enriched GO categories for blue and orange light

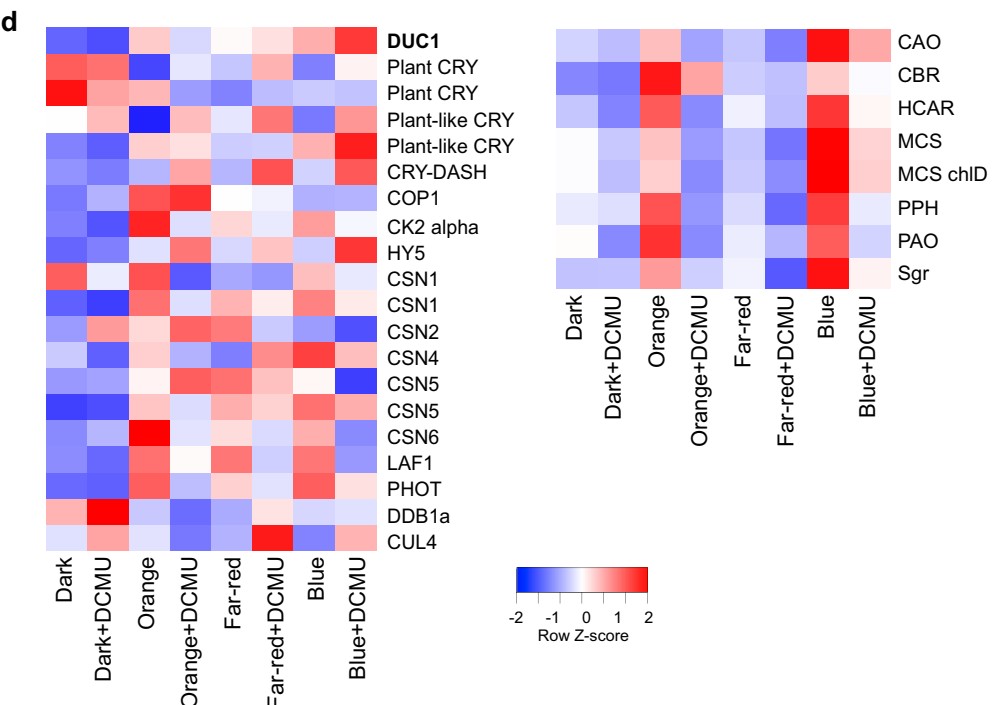

**Fig. 5 Transcriptome analysis under monochromatic blue, orange, and far-red light. a** Number of DEGs under each monochromatic light (blue, orange, far-red) against dark conditions. Right: samples were treated with DCMU; left: no DCMU treatment. The DEG is defined as >1.5-fold for gene expression with a *q*-value < 0.05. **b** Venn-diagram of number of DEGs in Fig. 5a. **c** GO enrichment analysis for 686 co-regulated DEGs under blue and orange light. **d** Expression values for DUC1 and light-signaling genes. The source data underlying Fig. 5b, d are provided as a Source Data file.

protect photosynthesis from high-light irradiation. Therefore, *P. provasolii* may adjust the amount of antenna complex by Chl degradation under strong light at the surface of the sea.

For photosynthesis in marine environments, Chl *b* is suitable because its absorption peak is shifted to blue-green compared with Chl *a*. This use of Chl *b* is different to land plants. Marine prasinophytes possess Chl *b*, not only in LHCs but also in PSI core antennae[5,32]. *P. provasolii* possesses LHCA, LHCB and a number of prasinophyte-specific LHCs (Lhcp) (Fig. 5). The Lhcps make complexes with Chl *a/b* and carotenoids[33]. Our transcriptome data showed that all *LHC* genes were upregulated and some *Lhcps* were strongly upregulated under orange- and blue-light irradiation (Fig. 5). This result is consistent with *LHC* regulation in land plants.

Our studies show the possible dynamic adaptation mechanisms to light spectra and intensities in *P. provasolii*, through the blue-, orange- and far-red-sensing DUC1. Acquisition of the dual functional photoreceptor, DUC1, has opened up a way for *P. provasolii* to widen its spectral utilization.

## Methods

**Prediction of *CRY* genes and identification of the host organism**. We used the assembled marine metagenome data sampled at Sendai Bay and the western subarctic Pacific Ocean 2012–2014 (http://marine-meta.healthscience.sci.waseda.ac.jp/crest/metacrest/graphs/) and checked the protein domains using InterProScan v5.40-77.0[34]. Proteins were defined as candidate CRYs if they had the following three domains: "DNA photolyase N-terminal (IPR006050)", "Rossman-like alpha/bata/alpha sandwich fold (IPR014729)", and "Cryptochrome/DNA photolyase, FAD-binding domain (IPR005101)", following the dbCRY method[35]. We also excluded known CRYs that matched with the NCBI nr database[36] by a blastp search[37]. To find the candidate host organism, we applied a blast search against MMETSP (The Marine Microbial Eukaryote Transcriptome Sequence Project) data[38]. The culture strain of *P. provasolii* NIES-2893 was obtained from the National Institute for Environmental Studies, Japan.

**Phylogenetic analyses of Phy and Cry regions of DUC1**. For phylogenetic analyses of DUC1, we divided it into Phy (1–3087 nucleotides) and Cry (3143–5073 nucleotides) regions. The datasets of Phy and Cry regions were composed of 54 operational taxonomic units (OTUs) and 96 OTUs, respectively, including available homologs from official databases. These sequences were aligned using mafft v7.453[39] with a linsi option, and ambiguous regions were trimmed using v1.4.rev15[40] with the option automated1. The trimmed dataset contained 457 and 353 amino acids for Phy and Cry regions, respectively. A model test was performed using ModelTest-NG v0.1.5[41]. Maximum likelihood analysis was performed using RAxML-NG v0.9.0[42] with 200 bootstrap replicates.

**Plasmid construction for spectral characterization of PpPHY and PpCRY**. The *Escherichia coli* strains Mach1 T1$^R$ (Thermo Fisher Scientific) and DH5α (TaKaRa) were used for DNA cloning. These cells were grown on Lysogeny Broth (LB) agar medium at 37 °C. The transformed cells were selected by 20 μg mL$^{-1}$ kanamycin or 100 μg mL$^{-1}$ ampicillin.

The *PpPHY* region (1–1986 bp in *PpDUC1*) was cloned into *Nde* I and *Bam*H I sites of the pET28a vector with N-terminal His-tag (Novagen) using the Gibson Assembly System (New England Biolabs, Japan). The DNA fragment of the *PpPHY* region was amplified by PCR using KOD One$^{TM}$ PCR Master Mix (Toyobo Life Science) with a codon-optimized synthetic gene for expression in *E. coli* (Genscript) and an appropriate nucleotide primer set (Supplementary Table 2). The pET28a vector was amplified by PCR using DNA polymerase with template DNA and an appropriate nucleotide primer set (Supplementary Table 2).

The *PpCRY* region (3115–5073 bp in *PpDUC1*) was cloned into the *EcoR* I and *Xba* I sites of a pCold GST DNA vector with an N-terminal GST-tag (TaKaRa) using restriction enzymes and ligase. A DNA fragment of the *PpCRY* region was amplified by PCR using PrimeSTAR Max DNA Polymerase (TaKaRa) with genomic DNA from *P. provasolii* and an appropriate nucleotide primer set (Supplementary Table 2). All of the plasmid constructs were verified by nucleotide sequencing (FASMAC).

**Expression and purification of PpPHY fused with His-tag**. The *E. coli* strain C41 (Cosmo Bio) was used for His-tagged PpPHY (amino acid positions 1–662 in PpDUC1) expression through the pKT270, pKT271 and pKT272 constructs as biliverdin IXα (BV), phycocyanobilin (PCB) and phytochromobilin (PΦB) synthetic systems, respectively[11,43]. Bacterial cells were grown in LB medium containing antibiotics (20 μg mL$^{-1}$ kanamycin and 20 μg mL$^{-1}$ chloramphenicol) at 37 °C. For protein expression, after the cells reached an optical density of 0.4–0.8 at

600 nm, isopropyl β-D-1-thiogalactopyranoside (IPTG) was added (final concentration, 0.1 mM), and the cells were cultured at 18 °C overnight.

After incubation, the culture broth was centrifuged at 5000 × *g* for 15 min to collect the cells. The cells were resuspended in lysis buffer A (20 mM HEPES–NaOH pH 7.5, 100 mM NaCl and 10% (w/v) glycerol) with 0.5 mM tris(2-carboxyethyl)phosphine, and then disrupted using an Emulsiflex C5 high-pressure homogenizer at 12,000 psi (Avestin). Homogenates were centrifuged at 165,000 × *g* for 30 min and then the supernatants were filtered through a 0.8 μm cellulose acetate membrane before loading on to a nickel-affinity HisTrap HP column (GE Healthcare) using the ÄKTA pure 25 (GE Healthcare) system. The column was washed using the lysis buffer containing 100 mM imidazole and, then, His-tagged PpPHY incorporated with each chromophore was eluted with a linear gradient of the lysis buffer containing 100 to 400 mM imidazole (1 mL min$^{-1}$, total 15 min). After incubation with 1 mM EDTA for 1 h, His-tagged PpPHY was dialyzed against the lysis buffer with 1 mM dithiothreitol (DTT) to remove EDTA and imidazole[44,45]. The purified proteins were dialyzed against lysis buffer A containing 1 mM dithiothreitol (DTT). Protein concentration was determined by the Bradford method.

**Expression and purification of PpCRY fused with GST-tag**. The *E. coli* strain C41 was used for GST-tagged PpCRY (amino acid positions 1039–1690 in PpDUC1) expression through the pG-KJE8 construct as a chaperone expression system (TaKaRa)[13]. Bacterial cells were grown in 8 L LB medium containing 500 μg mL$^{-1}$ L-arabinose and antibiotics (5 ng mL$^{-1}$ tetracycline, 100 μg mL$^{-1}$ ampicillin, and 20 μg mL$^{-1}$ chloramphenicol) at 37 °C. For protein expression, once the optical density at 600 nm of the cells reached 0.4–0.8, IPTG was added (final concentration, 1 mM), and the cells were cultured at 15 °C overnight.

After incubation, the culture broth was centrifuged at 5000 × *g* for 15 min to collect the cells. They were resuspended in lysis buffer B (20 mM HEPES–NaOH pH 7.5, 500 mM NaCl, 5 mM DTT and 10% (w/v) glycerol), and then disrupted using a homogenizer at 12,000 psi. The homogenate was centrifuged at 165,000 × *g* for 30 min and the supernatant filtered through a membrane before loading onto a glutathione-affinity GSTrap HP column (GE Healthcare). The column was washed with lysis buffer B to remove unbound proteins, and GST-tagged PpCRY was subsequently eluted with buffer containing 10 mM reduced glutathione. Protein concentration was determined by the Bradford method. The extracted protein was handled in the dark.

**Sodium dodecyl sulphate–polyacrylamide gel electrophoresis analysis for purified PpPHY and PpCRY**. Purified proteins were diluted into 60 mM DTT, 2% (w/v) SDS and 60 mM Tris-HCl, pH 8.0, and then denatured at 95 °C for 3 min. These samples were electrophoresed at room temperature using 10% (w/v) polyacrylamide gels with SDS. The gels were soaked in distilled water for 30 min followed by monitoring of fluorescence bands for detection of chromophores covalently bound to the proteins[46]. These bands were visualized through a 600-nm long-path filter upon excitation with blue ($\lambda_{max} = 470$ nm) and green ($\lambda_{max} = 527$ nm) light through a 562 nm short-path filter using WSE-6100 LuminoGraph (ATTO) and WSE-5500 VariRays (ATTO) machines. After the monitoring, the gels were stained with Coomassie Brilliant Blue R-250.

**UV–Vis spectroscopic analysis to monitor photocycles of PpPHY and PpCRY**. Ultraviolet and visible absorption spectra of the proteins were recorded with a UV-2600 spectrophotometer (SHIMADZU) at room temperature (r.t., approximately 20–25 °C). An Opto-Spectrum Generator (Hamamatsu Photonics, Inc.) was used to produce monochromatic light of various wavelengths to induce photoconversion.

**Assignment of the chromophores incorporated into PpPHY**. Sample solutions containing the native PpPHY in the dark state and the photoproduct state were diluted fivefold in 8 M acidic urea (pH < 2.0). The absorption spectra were recorded at r.t. before and after 3 min of illumination with white light. Assignment of the chromophores was conducted by comparing the spectra between the denatured PpPHY and standard proteins[47–49].

**Assignment of the chromophore incorporated into PpCRY**. Trichloroacetic acid was added to the sample solution containing the native PpCRY in the dark state to a final concentration of 440 mM. The solution was treated on ice for 1 h with shaking at 200 rpm. The precipitate was removed by centrifugation at 20,000 × *g* for 5 min, and then 50 μL of the supernatant were injected into a high-performance liquid chromatograph (Prominence HPLC system, Shimadzu). Released chromophore included in the supernatant was eluted isocratically (flow rate, 1 mL min$^{-1}$) with a solvent (MeOH/20 mM KH$_2$PO$_4$ = 20/80) and separated using a reverse-phase HPLC column (InertSustainSwift C18, 4.6 i.d. × 250 mm, 5 μm; GL Sciences) at 35 °C. The chromophore was detected by its absorption at 360 nm. Assignment of the chromophores was performed based on their retention times ($t_R$) compared with standard compounds; 5,10-methenyltetrahydrofolate chloride (MTHF chloride, Schircks Laboratories), flavin adenine dinucleotide (FAD, Tokyo Chemical Industry), flavin mononucleotide (FMN, Tokyo Chemical Industry) and riboflavin

(RF, Tokyo Chemical Industry). The sample and standard compounds were handled under dark conditions.

**Protein blot of PpDUC1-GFP and PpDUC1-HA expressed in plants**. Total protein was extracted from 80 mg of *A. thaliana* whole plants and *N. benthamiana* leaf discs using 240 μL of protein extraction buffer composing 50 mM Tris-HCl (pH 8.0), 150 mM NaCl, 0.1% 2-mercaptoethanol, 5% glycerol, 0.5% Triton X-100, and cOmplete™ protease inhibitor mini cocktail (Sigma). After grinding in the protein extraction buffer total slurry was centrifuged twice at 12,000 rpm for 5 min to remove precipitates. Ten microliters of supernatant was added with 5 μL of protein loading buffer and heat denatured. The denatured samples were loaded onto a 7.5% SDS-polyacrylamide gel for electrophoresis. After electrophoresis, proteins were electroblotted onto a polyvinylidene difluoride (PVDF) membrane (Millipore) in the blotting buffer composing 25 mM Tris, 192 mM glycine and 20% methanol. Subsequently, the membrane was incubated for 1.5 h in 1% of skimmed milk (Nakarai tesque), rinsed for 10 min twice with 1×TBST composing 137 mM NaCl, 2.68 mM KCl, 25 mM Tris-HCl, pH 7.4, 0.1 w/v% Tween-20. For detection of PpDUC1-HA, skim-milk blocked membrane was incubated with anti-HA peroxidase conjugate (1201381900, Roche 1:1000 dilution) for 1.5 h. For detection of PpDUC1-GFP, blocked membrane was incubated with rabbit anti-GFP antibody (TP401, Torrey Pines Biolabs Inc., 1:2000) for 20 h. The membrane was rinsed three times with 10 min intervals and incubated with protein A, horseradish peroxidase linked antibody (NA9120, GE Healthcare Corp., 1:2000) for 30 min. Both membranes were rinsed three times for 10 min each with 1×TBST and after incubation with enhanced chemiluminescence reagents (ECL Select™ Western Blotting Detection Reagents, GE Healthcare Corp.) They were processed with a chemi-luminescent image analyzer (Chem Doc XRS plus, BIO-RAD).

**Constructs for transient transformation in *N. benthamiana***. For infiltration into the leaf epidermal cells of *N. benthamiana*, the full-length open reading frames of *PpDUC1*, and parts of *PpPHY* (1–3621 bp in *DUC1*) and *PpCRY* (3115–5070 bp in *DUC1* without stop codon) were amplified from *P. provasolii* genomic DNA by PCR with each primer set (Supplementary Table 2) and cloned into the pDONR207 ENTRY vector by BP recombination using with BP Clonase II enzyme (Invitrogen Corp.). These open reading frames were transformed into the pEarleyGate 103 destination vector[50] to fuse in frame with GFP and 6× His by LR reaction. A synthetic gene of *Arabidopsis thaliana HY5* (At5g11260) fused to *mCherry* (Sequence ID: MH976504.1) was synthesized by Eurofins Genomics. The DNA fragment was amplified by PCR with primers (Supplementary Table 2) using the synthetic gene as a template and cloned into the pSK1 plasmid vector[51].

**Transient expression and observation of PpDUC1-GFP in *N. benthamiana* leaves**. Each construct was transformed into Agrobacterium GV3101. These agrobacteria were infiltrated into the leaf epidermal cells of *N. benthamiana* by eluting with 10 mM MES (pH 5.6) with 200 μM acetosyringone into the undersides of leaves of *N. benthamiana* plants[52]. A Zeiss LSM880 Airyscan Fast-mode microscope (Zeiss) with AxioObserver Z1 20X and W40X objectives was used to detect GFP fluorescence. The excitation wavelength was 488 nm, and a band-pass filter of 493 to 556 nm was used for emission for GFP. The excitation wavelength was 561 nm, and a band-pass filter of 578 to 640 nm was used for emission for mCherry.

**DNA extraction, genome sequencing, assembly, and annotation**. *P. provasolii* NIES-2893 cells were cultivated for 4–6 days in ~20 mL IMK medium (Nihon Pharmaceutical, Tokyo, Japan) under white LED light (~50 μmol photons m$^{-2}$ s$^{-1}$) with a 14 h:10 h light:dark cycle. DNA was extracted as described in Suzuki et al.[53] Cells were collected by gentle centrifugation and ground in a pre-cooled mortar with liquid nitrogen and 50 mg of 0.1 mm glass beads (Bertin, Rockville, MD, USA). The cells were incubated with 600 μL of CTAB extraction buffer[54] at 65 °C for 1 h. DNA was separated by mixing with 500 μL of chloroform and centrifuging at 20,000 × *g* for 1 min; it was concentrated by standard EtOH precipitation. The paired-end library was constructed using a TruSeq DNA PCR-free Kit (lllumina Inc., San Diego, CA, USA), according to the manufacturer's protocol. The mate-pair library was constructed using a Nextera Mate Pair Library Prep Kit (Illumina), according to the manufacturer's "gel-free" protocol. The paired-end and mate-pair libraries were sequenced using the MiSeq System (Illumina) with MiSeq Reagent Kits v3 (300 bp × 2) (Illumina). For nanopore sequencing, the library was constructed using a Rapid Sequencing Kit (SQK-RAD003, Oxford Nanopore Technologies, Oxford, UK). Sequencing was performed using a MinION with a SpotON Flow Cell (FLO-MIN106).

We sequenced 40,351,502 paired-end (9.7 Gbp), 10,652,248 mate-pair (1.7 Gbp), and 90,971 nanopore reads (0.50 Gbp, N50 = 18.8 Kbp). The raw reads were assembled into 47 scaffolds (22.9 Mbp) with one gap using MaSuRCA 3.3.2[55]. The nanopore reads were corrected using CONSENT v1.1.2[56]. The paired-end reads were corrected using Trimmomatic version 0.36[57]. The assembly was re-scaffolded by the corrected nanopore reads using Fast-SG[58] with the k-mer size = 70. Gaps were filled using LR_Gapcloser v1.1[59] with the corrected nanopore reads. Finally, we mapped the corrected paired-reads using BWA version 0.7.17[60], and polished the assembly using Pilon 1.22[61]. The polishing was performed twice.

We removed the two sequences of the complete chloroplast and mitochondrial genomes. Genome annotation was performed using the funannotate pipeline v1.5.3 (https://github.com/nextgenusfs/funannotate). For construction of the gene models, we used our RNA-seq reads (19 Gbp) under various light conditions (described in the transcriptome section) after read correction with Trimmomatic[57]. Functional annotation was performed using the eggNOG web server[62]. To check expression of the gene models, we mapped the RNA-seq reads used for the gene-model construction to the coding sequences (CDSs) using minimap2 2.17[63], and 99.5% and 98.4% of the predicted CDSs were mapped by at least one and five RNA-seq reads, respectively. The completeness of the gene prediction was assessed using BUSCO v2/v31 via gVolante[61].

**Transcriptome analysis under monochromatic light**. The cells of *P. provasolii* NIES-2893 were grown at 19.5 °C with shaking at 120 rpm in a light intensity of 13 μmol m$^{-2}$ s$^{-1}$ under a LD12:12 light/dark cycle. To monitor gene expression during irradiation of monochromatic light, cells were synchronized by a 12 h dark treatment before being exposed to monochromatic constant light. Light treatments were performed using LEDs (Nippon Medical & Chemical Instruments Co., Ltd., Japan) of specific wavelengths at an intensity of 7 μmol m$^{-2}$ s$^{-1}$. The following settings were used: for blue light, the peak was at 447 nm; for orange light, at 596 nm, and for far-red light, at 697 nm. The electron transport inhibitor 3-(3,4-dichlorophenyl)-1,1-dimethylurea (DCMU) was added to the samples before dark treatment. The cells were treated with 40 μM DCMU in 0.1% DMSO or with only 0.1% DMSO (control). After light treatment for 2 h with or without the addition of DCMU, cells were harvested and frozen with liquid nitrogen. After cell disruption using glass beads, total RNA was extracted from *P. provasolii* using the TRIzol reagent (Thermo Fisher Scientific Inc, USA). The extracted total RNA was purified using RNeasy Plant Mini Kits (QIAGEN, The Netherlands). Libraries for RNA-seq analysis were synthesized using Illumina® TruSeq® Stranded mRNA Sample Preparation Kits (Illumina Inc, USA) and sequenced on an Illumina HiSeq 2000 using directed paired-end technology. The sequenced reads were mapped to the *P. provasolii* genome with STAR v2.5.0c[62] after trimming the low-quality reads (Phred quality of ≤20) with the FASTX-Toolkit-0.0.14 (http://hannonlab.cshl.edu/fastx_toolkit/index.html). Read counts were normalized with DESeq 1.42.0 in an R package[63]. Differentially expressed genes (DEGs) were defined as >1.5-fold for gene expression with a *q*-value < 0.05.

**Reporting summary**. Further information on research design is available in the Nature Research Reporting Summary linked to this article.

## Data availability
Data supporting the findings of this work are available within the paper and its Supplementary Information files. A reporting summary for this Article is available as a Supplementary Information file. The datasets and plant materials generated and analyzed during the current study are available from the corresponding author upon request. The genome and transcriptome data are deposited in the DDBJ/EMBL/GenBank under the accession number of GCA_015473125.1, PRJDB10693, and PRJNA726377, respectively. The genome browser and transcriptome data of *P. provasolii* are available at: http://matsui-lab.riken.jp/JBrowse/index.html?data=data%2Fpycnococcus. The algal strains are available in the NIES collection: *P. provasolii* (NIES-2893) and *P. marina* (NIES-1419). The publicly available datasets in this study include: PhycoCosm database [https://phycocosm.jgi.doe.gov/phycocosm/home], Ocean Monitoring Database of Sendai Bay and the western subarctic Pacific Ocean [http://marine-meta.healthscience.sci.waseda.ac.jp/crest/metacrest/graphs/], and MMETSP (The Marine Microbial Eukaryote Transcriptome Sequence Project) [https://www.imicrobe.us/#/projects/104]. Source data are provided with this paper.

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

## Acknowledgements

This research was partially supported by the National BioResource Project (NBRP) from Japan Agency for Medical Research and Development (AMED). This research was supported by JSPS KAKENHI Grant Number 20K21438.

## Author contributions

Y.M. and A.S. identified the *DUC1* gene and analyzed its expression. S. Suzuki, H.Y., M. K. determined the *P. pravasolii* genome. K.F., R.N. confirmed the absorbance spectra of PpPHY and PpCRY. S. Shimada, M.H., T.K., and M.S. contributed the RNA analysis and complementation assay in *Arabidopsis*. H.H., E.O.K., and S. Shimada examined the intracellular localization of DUC1. K.Y., T.W., T.G., T.S. provided the metagenome data and the gDNA samples. Y.M., S. Suzuki, K.F., S. Shimada, Y.K., R.N., H.Y., M.K., M.M. wrote the manuscript.

## Competing interests

The authors declare no competing interests.
