## [Peer Review File · Nature Communications]

REVIEWER COMMENTS

Reviewer #1 (Remarks to the Author):

The paper reports the determination of the genome of the prasinophyte alga *Pyconococcus provasolii* sp. NIES-2893, focusing on the discovery of a novel chimeric photoreceptor named Dualchrome1 (PpDUC1) that was first identified in a marine metagenome survey. Also found in the genome of another related species, PpDUC1 encodes a prasinophyte phytochrome in which the C-terminal ATP-binding transmitter kinase and REC domains of phytochrome are replaced by a plant-type cryptochrome. The lack of introns in this gene suggests viral involvement in its origin only in species highly related to *P. provasolii*. Functional studies were presented to support the hypothesis that DUC1 performs a gene regulatory role involved in the light-regulation of photosynthesis-related genes in this organism. Since chimeric phytochromes have been identified in other algae and in some primitive land plants, this discovery is not paradigm shifting. Others includes neochrome, a fusion of a phytochrome photosensory region with a phototropin found in algae, hornworts and ferns, and fusions with ser/thr kinase domains found in selected mosses and other algae. Outlined below are the strengths and weaknesses of this paper. The weaknesses outweigh the strengths, so it is clear that substantial revision will be needed prior to publication of this work.

Major Strengths. The biochemical and spectroscopic characterization of the PpDUC1 photoreceptor is the major strength of this paper. This work is rigorous and carefully performed (Fig. 2, S2, S5, S6 and S7). The authors demonstrate that the photochemical properties of the phytochromes are similar to other prasinophyte PHYs. However, the photochemical properties of the PpCRY are atypical for a plant CRY since the photo-reduced flavin radical is not protonated and therefore cannot be affected by visible light. Whether the latter is an intrinsic property of plant-type prasinophyte CRYs would be interesting to follow up, i.e. there are two other plant-type CRYs in *P. provasolii*. The phylogenetic analysis of the PHY domain establish a prasinophyte origin for this part of PpDUC1. A more extensive phylogenetic analysis of the newly discovered CRY fusion domain might provide insight into the origin of this blue-light sensor would be interesting.

Major Weaknesses. The functional studies are one major weakness of this paper. First, the RNA-Seq analysis is superficial and incomplete. A complete list of annotated differentially regulated genes in Fig. 4A should be included and properly analyzed. Is there any evidence for light regulation of prasinoxanthin, Mg 2,4-D and/or the newly discovered chlorophyll degradation-related genes? Data presented is so limited that one cannot draw any conclusion about specific pathways regulated by light quality. The significance of differentially expressed genes (DEGs) in each regulatory class shown in Fig. 4A and B requires some to be confirmed by RT-PCR. The rudimentary photobiology protocol used also is insufficient to assign a function to PpDUC1 in gene regulation. Plant CRYs can absorb green orange and even red light, and opsin-related sensors are known that can sense orange light. Hence the two plant-type CRYs and carotenoid-based sensors could be responsible for the orange light response. The authors also need to rule out photosynthesis itself as the photoreceptor using DCMU treatment or by using PAR-adjusted fluence rate experiments at different wavelengths. Second, the lack of PpDUC1 complementation of the *Arabidopsis* phyB mutant requires determination whether the chimera was chromophorylated *in vivo*. Complementation of the *Arabidopsis* cry2 mutant also is not convincing and the complemented fluence rate response curves in Fig. S8B do not indicate complementation in the low fluence rate range. The rationale for performing complementation in the cry2 mutant background assumes the unlikely hypothesis that discrete functions of CRY1 and CRY2 have already 'separated' as far back as the prasinophyte lineage. For this reason, complementation studies in the cry1cry2 double mutant would be a far better experiment.

A second major weakness of this paper is the minimal analysis of the *P. provasolii* genome. This appears to be the first report of the genome of this species, and does not appear to do the genome justice by presenting much of its analysis in the Supplement. It might be more prudent to remove this aspect of the paper, i.e. Table 1, Figures S1 & S10-S23 and Tables S2-4 & S6-S11 could be removed,

and instead focus on the photoreceptor aspects of this work. Table S5 could be moved into the main paper to support a comparative analysis of photoreceptors present in chlorophytes. This would allow for some discussion about what is known about their expression and their properties in other prasinophytes such as *Micromonas pusilla*, e.g. see Duanmu et al 2014 Marine algae and land plants share conserved phytochrome signaling systems. PNAS 111, 15827. It is quite surprising that this paper was not discussed by the authors as it is quite relevant to their work. Aside from the genome analysis issue, the conclusion in the Abstract that the "complete genome revealed that *P. provasolii* facilitates light adaptation mechanisms via pheophorbide a oxygenase (Pao) and prasinoxanthin," is simply not supported by these investigations. PAO sequences are notoriously difficult to ascertain without biochemical support, and this reviewer cannot check this possibility since the protein sequences were not provided. The same is true for the putative SGR gene. SGR-related genes are found in many chlorophyte species, as well as in non-photosynthetic species. There also is no mention of chloroplast targeting sequences of these genes. Phylogenetic reconstructions of these two gene families are needed as a minimum to suggest that these genes are PAO or SGR orthologs.

Other concerns.

1. Sequence alignments for all phylogenetic reconstructions reported (Figs. 6, S3, S4 and S6B) should be provided as supplementary material.

2. Figs. 5 and 6 could be moved to the Supplement or omitted altogether since the former is not particularly novel (see Duanmu et al, 20154) and the latter trees are incomplete and not particularly informative.

3. Results

Line 116. 'photoconverter' might be replaced with 'photosensing'

Line 125-128. The authors need to cite literature for the ferredoxin-dependent bilin reductase family and/or present a phylogenetic tree that shows that the PpPCYA gene is found in the PCYA lineage.

Line 144 (and in Methods line 385). A citation to the pG-KJE8 plasmid is needed.

Line 182. Fig 3 shows that PpCRY and PpDUC are localized to discrete regions in cells. Confirmation is needed to show that these regions are indeed nuclei, e.g. with a DNA stain. These could be cytoplasmic aggregates or association with other structures. The title of this section needs to change otherwise.

Lines 187-189. You can only conclude that no other known photoreceptors absorb in the orange region.

Line 198. What does 'high completeness' mean? Details of genome analyses are insufficiently described and have been cherry-picked. Which genes were assigned using the transcriptome and which used with modeling? How were introns predicted? Could some genes have been missed?

4. Discussion

Section on Line 260. This discussion adds little to what has already been published on prasinophyte phytochromes.

Section beginning on line 278. This discussion repeats much of which is previously known. So much more could have been discussed, e.g. the interaction between PHY and CRY signaling, how orange, blue and far-red light might be integrated, etc. The repeated loss of PHY genes is more extensively discussed in the recent review by Rockwell et al, 2020

Section beginning on Line 293 "Dynamic adaptation .." This discussion repeats what is known from other photobiological studies, but this report provided no experimental connection between PpDUC1 function and dynamic light adaptation and the connection to chlorophyll degradation is overly speculative and without experimental support.

5. Methods

Lines 403-405. Was zinc added to visualize the bound bilin on gels? If so, citation needed.

Line 444. In ABRC, this Salk line is not characterized. ABRC online indicates that Salk line 022635 has an insertion in AT2G33060, a receptor kinase, not in PHYB. Is this correct?

Line 530. On what medium were cells plated?

5. Figures.

Fig 1D. *Volvox* and *Bathycoccus* are misspelled.

J. Clark Lagarias

Reviewer #2 (Remarks to the Author):

This interesting manuscript comprehensively describes the discovery of a novel chimeric photoreceptor: Dualchrome1 (DUC1). It is composed of a phytochrome and a cryptochrome. The authors identified it in marine metagenomes and were also able to link it back to the species which encoded it in its genome: The prasinophyte alga *Pycnococcus provasolii*. Its genome and transcriptome were sequenced in addition exploring the function of DUC1 by heterologous expression in *A. thaliana*. Expression of DUC1 in *E. coli* with subsequent biophysical characterization confirmed the predicted function.

Generally, I think this is an interesting discovery because it significantly adds to our understanding of the functional diversity of photoreceptors in the marine environment. The authors did a great job in combining descriptive sequence-led research with more mechanistic insights into the function and role of DUC1. Several different approaches were combined and the results all converged to confirm the function and potential biological role of this new photoreceptor. Thus, I am excited about these new results and the quality of this manuscript. Nevertheless, I have some suggestions to make to reduce some of the speculations in the manuscript and therefore to provide more rigor:

I did not see much evidence in the manuscript that *P. provasolii* is a cosmopolitan green alga (e.g. title). I also looked at the supplement. I think this can be easily addressed by exploring the TARA Oceans dataset (e.g. 18S and/or metagenomics). I think those data should feature even in the main manuscript (e.g. as part of figure 1) because they will strengthen the significance of DUC1.

To me, the paragraph on the evolutionary diversity (e.g. lines 278-291) can be improved by giving divergence estimates (e.g. figs 5, 6). For instance, a Bayesian Chain Monte Carlo (MCMC) analysis can be used for implementing it in BEAST (Bouckhaert et al. 2019). The results will give divergence estimates to substantiate the claims made in the manuscript.

The identification of Pao and Sgr is interesting. Can this be explored a little further in addition to improving the trees? For instance, any evidence for the role of horizontal gene transfer (Fig. 6a) in acquiring these genes?

I suggest to reverse the order for transcriptome profiling and genome sequencing. The genome should be discussed first.

Minor suggestions:

- 1) I don't think table 1 needs to be part of the main manuscript. Move it to the supplement. There is not much new information given anyway.
- 2) Add all bootstrap values to figure 1d.
- 3) I don't understand 'fold-change (Fc) <2/3 or FC > 1.5' in the figure legend for figure 4a. Please clarify. There should be one threshold and usually, it is ≥ 2 -fold for gene expression with a p-value ≤ 0.05 or lower.

Reviewer #3 (Remarks to the Author):

The description of the "dualchrome" (DUC1) photoreceptor discovered in this group of green algae is a significant contribution to our understanding of photoreceptor evolution and function, relating to both the phytochrome and cryptochrome groups of long-wavelength and short-wavelength visible light sensing molecules. The authors present convincing evidence that a chimeric structure consisting of the

N-terminal domains of a phytochrome-related structure ($\sim 3/4$ of a canonical phytochrome lacking the histidine kinase-related domain) fused to a large sequence related to the photolyase and FAD-binding domains of canonical cryptochromes is encoded in genomic and cDNA sequence of a prasinophyte alga, named *Pycnococcus provasolii*, identified from marine metagenomic sequencing projects. No separate canonical PHY gene is encoded in this organism but five additional individual CRY-related genes are encoded there. The authors have characterized the photochemical properties of in vitro expressed dualchrome apoprotein re-constituted with the presumed phycoyanobilin and FAD chromophores and conclude that it is a broad-spectrum sensor for UV-blue and orange-to-far red light. Previous studies have shown that reconstitution of algal PHY domains with bilin chromophores yields molecules that surprisingly can sense wavelengths across much of the visible spectrum. Nevertheless, discovery of this novel chimeric PHY-CRY dualchrome receptor identifies new structural possibilities for how such broadening of photochemical sensitivity can evolve. These findings also extend and expand the previous identification of what was called "neochrome" in ferns, a chimera of the PHY N-terminal region with phototropin LOV domain flavin-binding sequences. The evidence for complementation of the *Arabidopsis cry2* mutant phenotype by overexpressed dualchrome indicates that it is weakly functional in higher plant signaling. The cellular localization studies in tobacco cells indicate that it is nuclear-localized in the light. Although these experiments are clearly done in extremely non-native cell systems, the observations support in vivo functionality of the dualchrome protein. The dualchrome structure and its photochemical properties are novel, never having been described before, and constitute a significant advance in understanding how multiple photoreceptor protein/prosthetic-group modules have been genetically combined in diverse organisms to generate unique light-sensing molecular machinery.

One alteration I would suggest is to remove the last sections of the manuscript, from line 217 to line 234, and the corresponding data in Table 1 and Figure 6 and in Supplemental Figures S10-S23. These are extensive components of the paper that deal with descriptions of the genome, the light-harvesting complex component gene sequences, and the flagellar protein genes found in *P. provasolii*. These sections do not contribute to the core point of the paper, which is the discovery and analysis of dualchrome, and would logically be placed in a more extensive paper about the genome of this alga.

In addition, I do not think the work "unveiling" is appropriate in the title of a scientific paper. The word "Description" or "Identification" or just the title "A novel bifunctional photoreceptor, Dualchrome1, isolated from a cosmopolitan green alga" is more suitable.

In Figure S6, panel C, is the left panel in the Commassie and fluorescent gel photos supposed to be labeled PpPHY?

REVIEWER COMMENTS

Reviewer #1 (Remarks to the Author):

The paper reports the determination of the genome of the prasinophyte alga *Pyconococcus provasolii* sp. NIES-2893, focusing on the discovery of a novel chimeric photoreceptor named Dualchrome1 (PpDUC1) that was first identified in a marine metagenome survey. Also found in the genome of another related species, PpDUC1 encodes a prasinophyte phytochrome in which the C-terminal ATP-binding transmitter kinase and REC domains of phytochrome are replaced by a plant-type cryptochrome. The lack of introns in this gene suggests viral involvement in its origin only in species highly related to *P. provasolii*. Functional studies were presented to support the hypothesis that DUC1 performs a gene regulatory role involved in the light-regulation of photosynthesis-related genes in this organism. Since chimeric phytochromes have been identified in other algae and in some primitive land plants, this discovery is not paradigm shifting. Others includes neochrome, a fusion of a phytochrome photosensory region with a phototropin found in algae, hornworts and ferns, and fusions with ser/thr kinase domains found in selected mosses and other algae. Outlined below are the strengths and weaknesses of this paper. The weaknesses outweigh the strengths, so it is clear that substantial revision will be needed prior to publication of this work.

Major Strengths. The biochemical and spectroscopic characterization of the PpDUC1 photoreceptor is the major strength of this paper. This work is rigorous and carefully performed (Fig. 2, S2, S5, S6 and S7). The authors demonstrate that the photochemical properties of the phytochromes are similar to other prasinophyte PHYs. However, the photochemical properties of the PpCRY are atypical for a plant CRY since the photo-reduced flavin radical is not protonated and therefore cannot be affected by visible light. Whether the latter is an intrinsic property of plant-type prasinophyte CRYs would be interesting to follow up, i.e. there are two other plant-type CRYs in *P. provasolii*. The phylogenetic analysis of the PHY domain establish a prasinophyte origin for this part of PpDUC1. A more extensive phylogenetic analysis of the newly discovered CRY fusion domain might provide insight into the origin of this blue-light sensor would be interesting.

Major Weaknesses. The functional studies are one major weakness of this paper. First, the RNA-Seq analysis is superficial and incomplete. A complete list of annotated differentially regulated genes in Fig. 4A should be included and properly analyzed.

Reply: Following the reviewer's advice, we have carefully analyzed the RNA-seq data and presented them in Figs. 5, S9 and S11. The complete lists of the genes we present is in Table S3 and S4.

Is there any evidence for light regulation of prasinoxanthin, Mg 2,4-D and/or the newly discovered chlorophyll degradation-related genes?

Reply: The biosynthetic pathways of prasinoxanthin and Mg 2,4-D are not fully understood. Although we have checked the expression data of candidate prasinoxanthin precursors including lutein synthesis genes, they are not significantly induced by light. Since prasinoxanthin is important in low light conditions (Guillard, R. R. L., et al. 1991, Ref #5 in the manuscript), we assume that prasinoxanthin is also required under dark conditions. In this revised manuscript, we avoid any strong reference to this.

As for chlorophyll degradation-related genes including Pao, their expression pattern is similar to LHC genes. They are highly expressed in orange and blue light but not in orange light with DCMU treatment. These results have been updated in the manuscript.

Data presented is so limited that one cannot draw any conclusion about specific pathways regulated by light quality. The significance of differentially expressed genes (DEGs) in each regulatory class shown in Fig. 4A and B requires some to be confirmed by RT-PCR.

Reply: We have included experiments with DCMU for all the monochromatic lights. For the expression analysis, we examined LHC and DUC1 genes by real-time PCR. By comparing with RNA-Seq analysis, we confirmed that the RNA-Seq data coincides with the real-time experiments. From the DCMU experiment, we have been able to describe the effect of photosynthesis on gene expression especially in orange light. Interestingly, LHC expression was reduced by DCMU but that of ELIPs was increased. Also, DUC1, plant-type CRY homologs, HY5 and PHOT genes had increased expression with DCMU treatment. These observations indicate that genes involved in photosynthesis are directly controlled by photosynthetic activity. On the other hand, genes for light signal transduction are upregulated by DCMU treatment, indicating possible regulation of these light signal transduction genes.

The rudimentary photobiology protocol used also is insufficient to assign a function to PpDUC1 in gene regulation. Plant CRYs can absorb green orange and even red light, and opsin-related sensors are known that can sense orange light. Hence the two plant-type CRYs and carotenoid-based sensors could be responsible for the orange light response. The authors also need to rule out photosynthesis itself as the photoreceptor using DCMU treatment or by using PAR-adjusted fluence rate experiments at different wavelengths.

Reply: Related to the later comment, we agreed that we can only conclude that no other known photoreceptors absorb in the orange region in *P. provasolii*. We have carefully considered this point. Following this advice, we have performed RNA-Seq with and without DCMU. We observed that DCMU affected gene expression especially in orange light. There are several genes whose expression are not affected by DCMU treatment, indicating these are controlled by orange-light signal transduction. Most of the FR-induced gene expression was not changed by DCMU treatment. These genes may be under the control of the DUC1 receptor.

Second, the lack of PpDUC1 complementation of the Arabidopsis phyB mutant requires determination whether the chimera was chromophorylated in vivo. Complementation of the Arabidopsis cry2 mutant also is not convincing and the complemented fluence rate response curves in Fig. S8B do not indicate complementation in the low fluence rate range. The rationale for performing complementation in the cry2 mutant background assumes the unlikely hypothesis that discrete functions of CRY1 and CRY2 have already 'separated' as far back as the prasinophyte lineage. For this reason, complementation studies in the cry1cry2 double mutant would be a far better experiment.

Reply: Using a zinc assay, we have examined whether chromophore-bound PpDUC1 in a *phyB* mutant can be detected. We did not detect chromophore-bound PpDUC1-HA in the *Arabidopsis phyB* mutant even after anti-HA antibody concentration.

In vitro purified AtPHYB bound with PΦB protein prepared from *E. coli* can be detected by the zinc assay (A). We detected diluted AtPHYB bound with PΦB at as low as 50 ng of protein. This result indicates that a large amount of chromophore is needed for the zinc assay. We also could not detect chromophore-bound PpDUC1-GFP used in a transient assay in tobacco cells although transient assay usually expresses a higher amount of the transgene compared to transgenic plants (B).

We conclude that the amount of chromophore of PpDUC1-HA in the *phyB* mutant is too low to be detected by the zinc assay.

Detection of chromophore of PpDUC1 using zinc assay.

(A) Detection of a series of dilutions of *E. coli* expressing AtPHYB binding with PΦB. Each diluted recombinant AtPHYB-PΦB protein was loaded on to a 7.5% SDS-polyacrylamide gel. After electrophoresis, Zn²⁺-induced fluorescence of the holoprotein was visualized under UV irradiation after incubating the gel in a buffer containing 20 mM zinc acetate for 30 min.

(B) Detection of PpDUC1-GFP expressed in tobacco leaves. A total of 1.0 g of *N. benthamiana* leaf

material, which transiently expressed recombinant DUC1-GFP, was ground to a powder in liquid nitrogen and suspended in the following buffer: 50 mM Tris-HCl (pH8.0), 150 mM NaCl, 0.1% 2-mercaptoethanol, 5% glycerol, 0.5% triton -X-100, and cOmplete™ protease inhibitor mini cocktail (Sigma). The slurry was centrifuged twice to remove precipitates, concentrated and purified using a μMACS GFP Isolation Kit (Miltenyi Biotec K.K. Japan). 30 μl of the SDS denaturated supernatant were loaded onto a 7.5% SDS-polyacrylamide gel for electrophoresis.

We constructed *cry1cry2* with *PpDUC1*. We could not observe the clear complementation by *PpDUC1* in *cry1cry2*. The *cry1* effect on hypocotyl elongation is stronger compared to *cry2* and we could not observe clear differences under blue light between *cry1cry2* with and without *PpDUC1* transgene. We kept *cry2* data as supplemental information. We reexamined fluence-response again and expressed hypocotyl length as relative length compared to that of dark conditions. We moved these results to supplemental results.

A second major weakness of this paper is the minimal analysis of the *P. provasolii* genome. This appears to be the first report of the genome of this species, and does not appear to do the genome justice by presenting much of its analysis in the Supplement. It might be more prudent to remove this aspect of the paper, i.e. Table 1, Figures S1 & S10-S23 and Tables S2-4 & S6-S11 could be removed, and instead focus on the photoreceptor aspects of this work.

Reply: Following the reviewer's comment, we have rearranged our manuscript and removed Table 1, Figures S10, S11-S21 and Tables S2-4 & S6-S11. We have rewritten the text to focus on the photoreceptor aspects of the work.

Table S5 could be moved into the main paper to support a comparative analysis of photoreceptors present in chlorophytes. This would allow for some discussion about what is known about their expression and their properties in other prasinophytes such as *Micromonas pusilla*, e.g. see Duanmu et al 2014 Marine algae and land plants share conserved phytochrome signaling systems. PNAS 111, 15827. It is quite surprising that this paper was not discussed by the authors as it is quite relevant to their work.

Reply: Table S5 has been moved to the main text (Table 1 and the detailed gene names are in Table S2). We now discuss DUC1 in relation to other photoreceptors and signals of prasinophytes. We have also included the paper by Duanmu *et al.* in the discussion in this revised version.

Aside from the genome analysis issue, the conclusion in the Abstract that the "complete genome revealed that *P. provasolii* facilitates light adaptation mechanisms via pheophorbide a oxygenase (Pao) and prasinoxanthin," is simply not supported by these investigations. PAO sequences are notoriously difficult to ascertain without biochemical support, and this reviewer cannot check this possibility since the protein sequences were not provided.

Reply: We have removed the findings about PAO from the abstract, and the sentence has been rewritten as "Its complete genome suggests that *P. provasolii* facilitates light adaptation mechanisms". We have deposited these sequences in Genbank/DBJ with the

following IDs; PAO as GHP10940.1 and SGR as GHP05364.1.

The same is true for the putative SGR gene. SGR-related genes are found in many chlorophyte species, as well as in non-photosynthetic species. There also is no mention of chloroplast targeting sequences of these genes. Phylogenetic reconstructions of these two gene families are needed as a minimum to suggest that these genes are PAO or SGR orthologs.

Reply: This comment is related to the comment of reviewer #2. We reconstructed the phylogenetic trees for PAO and SGR orthologs (Fig. S9). As reported in Obata et al. (2019, MBE), SGR genes are conserved among non-photosynthetic bacteria and eukaryotic algae. However, cyanobacteria lack these genes. We inferred the SGR ortholog tree and got a similar result. *P. provasolii* possesses this gene and as do other prasinophytes. For PAO orthologs, we also re-inferred the PAO tree along with functional PAOs of land plants. This included plant PAOs identified in Tang et al. (2011, J. Plant. Physiol.), and these sequences were monophyletic with charophyceans (e.g., *Chlamydomonas*), *P. provasolii*, and cyanobacteria. Other prasinophytes, apart from *P. provasolii*, lack this gene, therefore, PAO may have been acquired from cyanobacteria via endosymbiotic gene transfer (EGT), resulting in many prasinophytes lacking the gene. Other paralogs of PAO, such as PTC52, CAO, CMO, and TIC55, evolved too rapidly to make an alignment with the *P. provasolii* PAO. Both SGR and PAO of *P. provasolii* are predicted to localize in the chloroplast using PredAlgo (Tardif et al. 2011, MBE).

Other concerns.

1. Sequence alignments for all phylogenetic reconstructions reported (Figs. 6, S3, S4 and S6B) should be provided as supplementary material.

Reply: We have added these multiple alignments to the supplementary data (Supp data 1).

2. Figs. 5 and 6 could be moved to the Supplement or omitted altogether since the former is not particularly novel (see Duanmu et al, 20154) and the latter trees are incomplete and not particularly informative.

Reply: We have re-calculated the phylogenetic tree and moved Fig. 6 to the supplemental data (Fig. S10).

Considering the evolutionary origin of DUC1, we think it is important to summarize the presence/absence of domain structure of PHY and pCRY based on the accurate phylogenetic tree which we constructed in Fig. 1. With this aspect, we brush up the phylogenetic tree and keep Fig. 5 (currently Fig. 4) as a main figure.

3. Results

Line 116. 'photoconverter' might be replaced with 'photosensing'

Reply: We have replaced 'photoconverter' with 'photosensing'.

Line 125-128. The authors need to cite literature for the ferredoxin-dependent bilin reductase family and/or present a phylogenetic tree that shows that the PpPCYA gene is found in the PCYA lineage.

Reply: We have cited literature for the ferredoxin-dependent bilin reductase family and presented a phylogenetic tree showing the PpPCYA gene in the PCYA lineage. (Frankenberg, N., Mukougawa, K., Kohchi, T. & Lagarias, J. C. Functional genomic analysis of the HY2 family of ferredoxin-dependent bilin reductases from oxygenic photosynthetic organisms. *Plant Cell* 13, 965–978 (2001).)

Line 144 (and in Methods line 385). A citation to the pG-KJE8 plasmid is needed.

Reply: We have added a citation to the pG-KJE8 plasmid.

Line 182. Fig 3 shows that PpCRY and PpDUC are localized to discrete regions in cells. Confirmation is needed to show that these regions are indeed nuclei, e.g. with a DNA stain. These could be cytoplasmic aggregates or association with other structures. The title of this section needs to change otherwise.

Reply: We have used HY5::mCherry to highlight the nucleus and indicated the localization of PpCRY and PpDUC in both the nucleus and the cytoplasm. We have also changed the title of this section.

Lines 187-189. You can only conclude that no other known photoreceptors absorb in the orange region.

Reply: We understand the point and describe the possibility of absorption of orange light by other photoreceptors.

Line 198. What does 'high completeness' mean? Details of genome analyses are insufficiently described and have been cherry-picked. Which genes were assigned using the transcriptome and which used with modeling? How were introns predicted? Could some genes have been missed?

Reply: We have removed the description of “high completeness”. We have checked genome completeness by detecting lineage-specific single-copy genes using BUSCO via gVolante (<https://gvolante.riken.jp/>), and the completeness of this genome was predicted at least 89.1%. For gene-model prediction, we used RNA-seq data under monochromatic light conditions (DDBJ BioProject ID: PRJDB10693, 12 samples, in total 19 Gbp data). The intron prediction (i.e. gene-model prediction) was performed using the Funannotate pipeline (<https://github.com/nextgenusfs/funannotate>), which performed *ab initio* and evidence-based gene model construction using RNA-seq and conserved proteins. 99.5% and 98.4% of the predicted CDSs were mapped by at least one and five RNA-seq reads, respectively. We added these expression rates in the manuscript.

4. Discussion

Section on Line 260. This discussion adds little to what has already been published on prasinophyte phytochromes. Section beginning on line 278. This discussion repeats much of

which is previously known.

Reply: We fully agree with your opinion, and have deleted the descriptions overlapped with the previous papers. Instead, we have kept the original discussion focusing on Tyr positions for blue-shifting.

So much more could have been discussed, e.g. the interaction between PHY and CRY signaling, how orange, blue and far-red light might be integrated, etc.

Reply: Following this advice, we discuss the interaction of PHY and CRY signaling including physical interaction of these two receptors in Arabidopsis. We also discuss the possibility of orange, blue and far-red light signal interaction in the discussion section of "Possible function of PpDUC1".

The repeated loss of PHY genes is more extensively discussed in the recent review by Rockwell et al, 2020

Reply: In this manuscript, we particularly focused on DUC1 evolution (i.e., evolution of phytochrome and plant cryptochrome) in early diverging Viridiplantae. Compared to the previous reports such as Rockwell *et al.* (2020), our study has two advantages: it focuses on the main lineages of early diverging groups of Viridiplantae (especially those formerly identified as "prasinophytes") and discusses the evolution based on highly-resolved phylogenetic relationships (Fig. 1C).

Section beginning on Line 293 "Dynamic adaption .." This discussion repeats what is known from other photobiological studies, but this report provided no experimental connection between PpDUC1 function and dynamic light adaptation and the connection to chlorophyll degradation is overly speculative and without experimental support.

Reply: We have rewritten the possible functions of DUC1 in response to different light signals. We describe chlorophyll degradation separately.

5. Methods

Lines 403-405. Was zinc added to visualize the bound bilin on gels? If so, citation needed.

Reply: Yes, we have added the following citation: Berkelman, T. R. & Clark Lagarias, J. Visualization of bilin-linked peptides and proteins in polyacrylamide gels. *Analytical Biochemistry* vol. 156 194–201 (1986).

Line 444. In ABRC, this Salk line is not characterized. ABRC online indicates that Salk line 022635 has an insertion in AT2G33060, a receptor kinase, not in PHYB. Is this correct?

Reply: The Salk number was a mistake. We have corrected it.

Line 530. On what medium were cells plated?

Reply: This was our mistake. We have removed “plated”.

5. Figures.

Fig 1D. Volvox and Bathycoccus are misspelled.

Reply: We have corrected the spellings.

J. Clark Lagarias

Reviewer #2 (Remarks to the Author):

This interesting manuscript comprehensively describes the discovery of a novel chimeric photoreceptor: Dualchrome1 (DUC1). It is composed of a phytochrome and a cryptochrome. The authors identified it in marine metagenomes and were also able to link it back to the species which encoded it in its genome: The prasinophyte alga *Pycnococcus provasolii*. Its genome and transcriptome were sequenced in addition exploring the function of DUC1 by heterologous expression in *A. thaliana*. Expression of DUC1 in *E. coli* with subsequent biophysical characterization confirmed the predicted function.

Generally, I think this is an interesting discovery because it significantly adds to our understanding of the functional diversity of photoreceptors in the marine environment. The authors did a great job in combining descriptive sequence-led research with more mechanistic insights into the function and role of DUC1. Several different approaches were combined and the results all converged to confirm the function and potential biological role of this new photoreceptor. Thus, I am excited about these new results and the quality of this manuscript. Nevertheless, I have some suggestions to make to reduce some of the speculations in the manuscript and therefore to provide more rigor:

I did not see much evidence in the manuscript that *P. provasolii* is a cosmopolitan green alga (e.g. title).

I also looked at the supplement. I think this can be easily addressed by exploring the TARA Oceans dataset (e.g. 18S and/or metagenomics). I think those data should feature even in the main manuscript (e.g. as part of figure 1) because they will strengthen the significance of DUC1.

Reply: Tragin M., et al. (Sci Rep. 2018 Sep 19;8(1):14020.) suggested that the order Pseudoscourfieldiales, which includes *Pycnococcus*, was observed in around 20% of their sampling stations. Also, Pseudoscourfieldiales had more than 1% of the Chlorophyta reads at only two stations. However, our metagenomic analysis showed high diversity of *P. provasolii* in the western subarctic Pacific Ocean in specific seasons, suggesting the ecological importance of this species in the ocean. In the manuscript, we have changed “cosmopolitan” to “oceanic green picoplankton”.

To me, the paragraph on the evolutionary diversity (e.g. lines 278-291) can be improved by giving divergence estimates (e.g. figs 5, 6). For instance, a Bayesian Chain Monte Carlo (MCMC) analysis can be used for implementing it in BEAST (Bouckhaert et al. 2019). The results will give divergence estimates to substantiate the claims made in the manuscript.

Reply: Thank you for your suggestion. To improve the reliability of the evolutionary history of phytochrome (phy) and plant-type cryptochrome (pCry), we re-inferred the phylogenetic analyses with more taxon sampling and Bayesian analysis using MrBayes. Please see Fig. 1C. The tree topology was backed up with high supported values (BP/BPP) and is similar to a recent report (Li et al. 2020, Nat. Ecol. Evol.). Based on this topology, we can simply discuss the evolutionary history (Fig.5). In particular, *P. provasolii* is sister to Nephroselmis, which possesses both of phy and pCry, strongly suggesting that the gene fusion of Duc1 had occurred in the Pycnococcus lineage after the branching of Nephroselmis. We attempted to estimate the ancestral form of the presence/absence of phy and pCry based on MCMC. However, it was difficult to interpret the results because the analyses do not seem to consider evolutionary direction. Based on the phylogenetic trees of phy and pCry of the green algae, they are monophyletic and coincide with the tree topology of the species, suggesting that the genes are unlikely to have been acquired via horizontal gene transfer. Therefore, we considered only the deletion of phy/pCry. For Fig. 6, we have also re-inferred the phylogenetic trees with Bayesian analyses. Please also see our next reply.

The identification of Pao and Sgr is interesting. Can this be explored a little further in addition to improving the trees? For instance, any evidence for the role of horizontal gene transfer (Fig. 6a) in acquiring these genes?

Reply: This comment is related to the comment of reviewer #1. We have reconstructed the phylogenetic trees for PAO and SGR orthologs. As reported in Obata *et al.* (2019, MBE), SGR genes are conserved among non-photosynthetic bacteria and eukaryotic algae. However, cyanobacteria lack these genes. We inferred the SGR ortholog tree and got a similar result. *P. provasolii* possesses this gene and as do other prasinophytes. For PAO orthologs, we also re-inferred the PAO tree along with functional proteins of land plants. This included plant PAOs identified in Tang *et al.* (2011, J. Plant. Physiol.), and these sequences were monophyletic with charophyceans (e.g., *Chlamydomonas*), *P. provasolii*, and cyanobacteria. Other prasinophytes, apart from *P. provasolii*, lack this gene, therefore, PAO may have been acquired from cyanobacteria via endosymbiotic gene transfer (EGT), and then many prasinophytes would lack the gene. Both SGR and PAO of *P. provasolii* are predicted to localize in the chloroplast using PredAlgo (Tardif et al. 2011, MBE).

I suggest to reverse the order for transcriptome profiling and genome sequencing. The genome should be discussed first.

Reply: We have changed the order of the descriptions of the transcriptome profiling and genome sequencing, discussing the genome first.

Minor suggestions:

1) I don't think table 1 needs to be part of the main manuscript. Move it to the supplement. There is not much new information given anyway.

Reply: We have made Table 1 part of Fig. 1.

2) Add all bootstrap values to figure 1d.

Reply: We have added bootstrap percentages (BPs) and Bayesian posterior probabilities (BPP) to Figure 1D. Instead of adding "100/0.01" to many lines, we indicate them with bold lines.

3) I don't understand 'fold-change (Fc) <2/3 or FC > 1.5' in the figure legend for figure 4a. Please clarify. There should be one threshold and usually, it is ≥ 2 -fold for gene expression with a p-value ≤ 0.05 or lower.

Reply: We have changed this to "The DEG is defined as > 1.5-fold for gene expression with a q-value < 0.05." We used q-value, not p-value, to avoid the multiple testing problem. The threshold of q-value becomes higher than the same value of p-value.

Reviewer #3 (Remarks to the Author):

The description of the "dualchrome" (DUC1) photoreceptor discovered in this group of green algae is a significant contribution to our understanding of photoreceptor evolution and function, relating to both the phytochrome and cryptochrome groups of long-wavelength and short-wavelength visible light sensing molecules. The authors present convincing evidence that a chimeric structure consisting of the N-terminal domains of a phytochrome-related structure (~3/4 of a canonical phytochrome lacking the histidine kinase-related domain) fused to a large sequence related to the photolyase and FAD-binding domains of canonical cryptochromes is encoded in genomic and cDNA sequence of a prasinophyte alga, named *Pycnococcus provasolii*, identified from marine metagenomic sequencing projects. No separate canonical PHY gene is encoded in this organism but five additional individual CRY-related genes are encoded there. The authors have characterized the photochemical properties of in vitro expressed dualchrome apoprotein re-constituted with the presumed phycocyanobilin and FAD chromophores and conclude that it is a broad-spectrum sensor for UV-blue and orange-to-far red light. Previous studies have shown that reconstitution of algal PHY domains with bilin chromophores yields molecules that surprisingly can sense wavelengths across much of the visible spectrum. Nevertheless, discovery of this novel chimeric PHY-CRY dualchrome receptor identifies new structural possibilities for how such broadening of photochemical sensitivity can evolve. These findings also extend and expand the previous identification of what was called "neochrome" in ferns, a chimera of the PHY N-terminal region with phototropin LOV domain flavin-binding sequences. The evidence for complementation of the *Arabidopsis cry2* mutant phenotype by overexpressed dualchrome indicates that it is weakly functional in higher plant signaling. The cellular localization studies in tobacco cells indicate that it is nuclear-localized in the light. Although these experiments are clearly done in extremely non-native

cell systems, the observations support in vivo functionality of the dualchrome protein. The dualchrome structure and its photochemical properties are novel, never having been described before, and constitute a significant advance in understanding how multiple photoreceptor protein/prosthetic-group modules have been genetically combined in diverse organisms to generate unique light-sensing molecular machinery.

One alteration I would suggest is to remove the last sections of the manuscript, from line 217 to line 234, and the corresponding data in Table 1 and Figure 6 and in Supplemental Figures S10-S23. These are extensive components of the paper that deal with descriptions of the genome, the light-harvesting complex component gene sequences, and the flagellar protein genes found in *P. provasolii*. These sections do not contribute to the core point of the paper, which is the discovery and analysis of dualchrome, and would logically be placed in a more extensive paper about the genome of this alga.

Reply: We have removed the last sections of the manuscript, from line 217 to line 234, and the corresponding data in Table 1 and Figure 6 and in Supplemental Figures S10-S23 dealing with the genome of *P. provasolii*. We have rewritten the manuscript according to these changes.

In addition, I do not think the work “unveiling” is appropriate in the title of a scientific paper. The word “Description” or “Identification” or just the title “A novel bifunctional photoreceptor, Dualchrome1, isolated from a cosmopolitan green alga” is more suitable.

Reply: We have renamed the manuscript as “A novel bifunctional photoreceptor, Dualchrome1, isolated from an oceanic green picoplankton”.

In Figure S6, panel C, is the left panel in the Commassie and fluorescent gel photos supposed to be labeled PpPHY?

Reply: Yes. We have fixed the label to PpPHY.

REVIEWERS' COMMENTS

Reviewer #1 (Remarks to the Author):

The authors have significantly improved the paper by focusing on the pDUC1 receptor. In particular, the RNA-Seq revisions and RT-PCR additions are welcome changes. Also welcome are the changes to the phylogenetic analyses, which are now more rigorous. The manuscript still falls short of establishing a function to pDUC1 in *P. provasolii*, but the evidence is convincing that both PHY and CRY components are photochemically active light sensors of prasinophyte providence. Key photobiological, e.g. O/FR reversibility, and genetic experiments that address this interesting issue remain for the future. The Discussion of the findings are appropriately more nuanced in this regard. I still have a few comments (outline below), which the authors may wish to address. However, in light of the changes made in response to the reviewer's comments, my additional concerns are minor which should not preclude publication.

Other comments.

1. Line 108. I think citation to Fig. 1C should be 1B.
2. Line 159. Which *Drosophila* dCRY is *Drosophila melanogaster* O77059 or *Drosophila melanogaster* Q24281 shown in Fig. S3? Maybe you should add the dCRY label in Fig. S3
3. Line 160. Is CraCRY the same as *Chlamydomonas reinhardtii* CR06G09700 in Fig. S3? Maybe should include the CraCRY label in Fig. S3?
4. Line 168. Fix spelling of *Micromonas* and add CCMP1545 to avoid species confusion.
5. Lines 175-176. In Fig. 3C looks like PpDUC1 is mostly in the nucleus. How was localization in cytoplasm shown? GFP control shows a lot of GFP accumulating in the nucleus with much more evidence for cytosolic localization. Not sure that heterologous expression experiments are informative here since interaction between PpPHY/CRY/DUC1 with tobacco proteins may be unnatural and lead to mislocalization. It is interesting that PpDUC1 exhibits 'mostly' nuclear localization since heterologous expression of other cryptogam phytochromes in plants leads to cytosolic localization predominantly.
6. Lines 210-217. This reviewer is still not convinced that these genes encode proteins that have the activities suggested. For this reason, the authors might consider downplaying this and to remove Fig. S10 and text from lines 210-217.
7. Line 215. The lack of an RCCR gene does not support the conclusion that *P. provasolii* can degrade Chl a via RCC.
8. Line 320. Please define Mg 2,4-D for general readers.
9. Line 687. Clark should be omitted and initial C. added after J.
10. Fig. S1. The colors chosen are hard to distinguish.
11. New Fig. 4 Much improved. The authors might consider switching the order of clades for better phylogenetic presentation. From bottom: glaucophyta, cryptophyta, chlorophyta, streptophyta then Bacteria and fungi (at top)

J. Clark Lagarias

Reviewer #2 (Remarks to the Author):

Many thanks for addressing all of my suggestions. Your manuscript has significantly improved. I have no additional suggestions to make. Well done.

Thomas Mock

Reviewer #3 (Remarks to the Author):

The authors have addressed my concerns about the paper. I think the extensive comments by other reviewers were on-target with regard to the biochemistry and molecular genetics of algal photoreceptors. The author's responses to those concerns/perceived weaknesses have improved the paper considerably.

REVIEWER COMMENTS

Reviewer #1 (Remarks to the Author):

The authors have significantly improved the paper by focusing on the pDUC1 receptor. In particular, the RNA-Seq revisions and RT-PCR additions are welcome changes. Also welcome are the changes to the phylogenetic analyses, which are now more rigorous. The manuscript still falls short of establishing a function to pDUC1 in *P. provasolii*, but the evidence is convincing that both PHY and CRY components are photochemically active light sensors of prasinophyte providence. Key photobiological, e.g. O/FR reversibility, and genetic experiments that address this interesting issue remain for the future. The Discussion of the findings are appropriately more nuanced in this regard. I still have a few comments (outline below), which the authors may wish to address. However, in light of the changes made in response to the reviewer's comments, my additional concerns are minor which should not preclude publication.

Other comments.

1. Line 108. I think citation to Fig. 1C should be 1B.

Reply: We have corrected it to Fig. 1b.

2. Line 159. Which *Drosophila* dCRY is *Drosophila melanogaster* O77059 or *Drosophila melanogaster* Q24281 shown in Fig. S3? Maybe you should add the dCRY label in Fig. S3

Reply: We show both O77059 *Drosophila melanogaster* O77059 and *Drosophila melanogaster* Q24281 in Fig. S3. Following the advice, we have added the dCRY label to O77059.

3. Line 160. Is CraCRY the same as *Chlamydomonas reinhardtii* CR06G09700 in Fig. S3? Maybe should include the CraCRY label in Fig. S3?

Reply: CraCRY is the same as the homolog in the clade, "Animal-like Cry, CPF1, and (6-4) photolyase" in Fig. S3. We have added CraCRY label in Fig. S3.

4. Line 168. Fix spelling of *Micromonas* and add CCMP1545 to avoid species confusion.

Reply: We have corrected the spelling and added the strain name.

5. Lines 175-176. In Fig. 3C looks like PpDUC1 is mostly in the nucleus. How was localization in cytoplasm shown? GFP control shows a lot of GFP accumulating in the nucleus with much more evidence for cytosolic localization. Not sure that heterologous expression experiments are informative here since interaction between PpPHY/CRY/DUC1 with tobacco proteins may be unnatural and lead to mislocalization. It is interesting that PpDUC1 exhibits 'mostly' nuclear localization since heterologous expression of other cryptogam phytochromes in plants leads to cytosolic localization predominantly.

Reply: As you suggested we speculated cytoplasm localization from fluorescence by GFP. GFP localizes both nucleus and cytoplasm. We need to understand how light signals perceived by DUC1 are transduced in *P. provasolii*. In this heterologous tobacco experiment we could observe PpPHY is not localised in nucleus but PpCRY and DUC1 localized in "mostly" in nucleus. We described that this observation is a heterologous system in the discussion and also we have corrected the description of subcellular localization of PpDUC1 that it is "mostly" in the nucleus.

6. Lines 210-217. This reviewer is still not convinced that these genes encode proteins that have the activities suggested. For this reason, the authors might consider downplaying this and to remove Fig. S10 and text from lines 210-217.

Reply: As the reviewer #1 suggested, we removed L210-217 and Fig. S10. We also removed the related parts in discussion (L.321-325).

7. Line 215. The lack of an RCCR gene does not support the conclusion that *P. provasolii* can

degrade Chl a via RCC.

Reply: As we described above (reply to the comment #6), we removed this paragraph in the text.

8. Line 320. Please define Mg 2,4-D for general readers.

Reply: We spelled out Mg 2,4-D to “Magnesium 2,4-divinylpheoporphyrin a_5 monomethyl ester”.

9. Line 687. Clark should be omitted and initial C. added after J.

Reply: We have corrected the author’s name to: Lagarias, J. C.

10. Fig. S1. The colors chosen are hard to distinguish.

Reply: We have changed colours with thin black lines in the Fig. S1. Especially, we emphasized *P. provasolii* with yellow.

11. New Fig. 4 Much improved. The authors might consider switching the order of clades for better phylogenetic presentation. From bottom: glaucophyta, cryptophyta, chlorophyta, streptophyta then Bacteria and fungi (at top)

Reply: We have changed the order of groups in Fig. S4 as the reviewer #1 suggested.

J. Clark Lagarias

Reviewer #2 (Remarks to the Author):

Many thanks for addressing all of my suggestions. Your manuscript has significantly improved. I have no additional suggestions to make. Well done.

Thomas Mock

Reviewer #3 (Remarks to the Author):

The authors have addressed my concerns about the paper. I think the extensive comments by other reviewers were on-target with regard to the biochemistry and molecular genetics of algal photoreceptors. The author’s responses to those concerns/perceived weaknesses have improved the paper considerably.